# Solution of master equations by fermionic-duality: Time-dependent charge and heat currents through an interacting quantum dot proximized by a superconductor

Lara C. Ortmanns[1], Maarten R. Wegewijs[2] and Janine Splettstoesser[3]

**1** Institute for Theory of Statistical Physics, RWTH Aachen, 52056 Aachen, Germany
**2** JARA-FIT, 52056 Aachen, Germany
**3** Department of Microtechnology and Nanoscience (MC2), Chalmers University of Technology, SE-41296 Göteborg

## Abstract

We analyze the time-dependent solution of master equations by exploiting fermionic duality, a dissipative symmetry applicable to a large class of open systems describing quantum transport. Whereas previous studies mostly exploited duality relations after partially solving the evolution equations, we here systematically exploit the invariance under the fermionic duality mapping from the very beginning when setting up these equations. Moreover, we extend the resulting simplifications –so far applied to the local state evolution– to non-local observables such as transport currents. We showcase the exploitation of fermionic duality for a quantum dot with strong interaction –covering both the repulsive and attractive case– proximized by contact with a large-gap superconductor which is weakly probed by charge and heat currents into a wide-band normal-metal electrode. We derive the complete time-dependent analytical solution of this problem involving non-equilibrium Cooper pair transport, Andreev bound states and strong interaction. Additionally exploiting detailed balance we show that even for this relatively complex problem the evolution towards the stationary state can be understood analytically in terms of the stationary state of the system itself via its relation to the stationary state of a dual system with inverted Coulomb interaction, superconducting pairing and applied voltages.

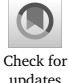

# 1  Introduction

The solution of dynamical equations of open quantum systems exhibiting many-body effects is inherently a more difficult problem than for closed systems. One reason for this is the reduction of the number of symmetries, due to the contact with the macroscopic, dissipative environment. As a result, it may seem that in general there is no systematic way to simplify the solution procedure and the physical analysis of the results of interest. This problem persists even for weakly coupled systems addressed in prior works on dissipative symmetries [1,2] and even when symmetries like detailed balance can be exploited [3–8].

Recently, it has been found that *fermionic* open systems, even when strongly coupled to wide-band metallic contacts, exhibit an extremely useful "symmetry" that is quite general. This so-called fermionic duality relation is truly dissipative in nature due to the explicit involvement of the special fermion-parity decay rate $\Gamma$ depending only on the interface properties [9, 10]. Using the quantum master equation description of the dynamics, $\partial_t \rho(t) = -i[H_S, \rho(t)] + \int_0^t ds \mathcal{W}(t-s)\rho(s)$, this duality in its most general form [9, 10] can

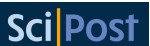

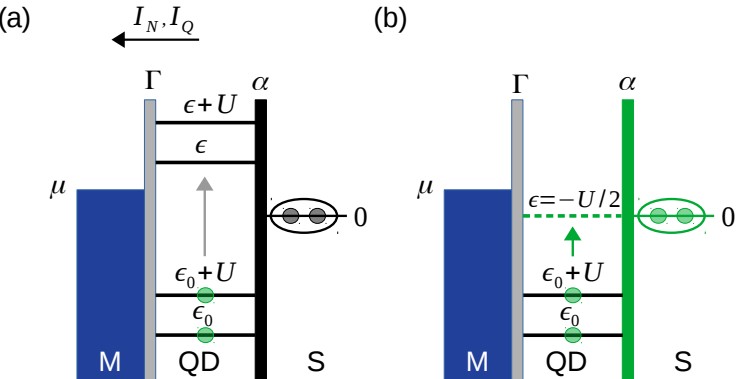

Figure 1: Single-level quantum dot with interaction $U$ and pairing $\alpha$ due to proximity of a superconducting contact with a large gap (infinite, not indicated). The dot is probed by charge- and heat-currents into a weakly coupled metal M. Of interest are the indicated transient charge and heat currents when the dot is initialized, for example, by a change of gate-voltage to a value (a) far from or (b) close to the superconducting resonance. Here we consider any given initial state that is a mixed state in the energy eigenbasis [Eq. (19)], independent of the details of its preparation considered elsewhere [19].

be expressed in terms of the frequency-domain memory kernel $W(\omega) = \int_0^\infty dt\, e^{i\omega t}\mathcal{W}(t)$ as follows: $W(\omega) = -\Gamma\mathcal{I} + \mathcal{P}\bar{W}(i\Gamma - \omega^*)^\dagger\mathcal{P}$. It relates the kernel matrix to its adjoint, conjugated with fermion parity $\mathcal{P}$ and shifted by a scalar $\Gamma$ and likewise shifts and mirrors the Laplace frequency $\omega$. This is a very strong restriction on the spectral properties of $W$ which determine the dynamics of the density operator $\rho(t)$. Postponing the details and specifics, we highlight that unlike ordinary symmetries this relation additionally involves a transformation of system *parameters* indicated by overbar ¯ in addition to its dynamical variables. Fermionic duality is valid for weak bilinear energy-independent coupling to a metal [9, 11–15] but has been extended to energy-dependent coupling [16] and combined with detailed balance [16]. As signalled by $\mathcal{P}$ and †, fermionic duality turns out to include the so-called PT-symmetry specific to Markovian Lindblad dynamics [2] which it generalizes to strongly coupled non-Markovian systems. Indeed, duality as formulated above was shown to remain valid [9, 10] and was exploited [17, 18] for strong bilinear but energy-independent coupling and parity-conserving quantum-dot Hamiltonians $H_S$.

Fermionic duality has been shown to facilitate stationary as well as time-dependent solutions of master equations (occupations) and quantum master equations (including coherences). Most importantly, it yields unexpected insights into the physical parameter dependence of both the time scales and the *amplitudes* of the state of a decaying quantum system. These affect local observable quantities such as charge and energy and their non-local transport currents. The power of the constraints imposed by fermionic duality was illustrated for relatively simple cases where it completely dictates –rather than just restricts– the form of the master equation [12]. Such constraints are extremely valuable for more complicated systems of practical interest for which an intuitive analysis of analytical results is essentially ruled out whenever various physical effects stand in competition. Indeed, such an analysis of the problem solved here has motivated the present work [19].

In the above cited prior works, fermionic duality was applied concretely to the solution of the master-equation eigenvalue problem by expressing half of the eigenvectors of $W$ in relation to the other half, which is assumed to be known by some standard calculation. It was not systematically exploited to find suitable variables and a suitable basis that would simplify

finding the first half of the solution, and, more importantly, to automatically bring it into a form that facilitates its analysis. In this paper we push this idea of fermionic duality as "generalized hermicity" further: going beyond Ref. [9], we consider from the very beginning quantities that are invariant under the duality transformation in a weakly coupled open quantum system and refer to these as *duality invariants*. Moreover, we apply this analysis not only to the transition rates (that govern the evolution of a state via the quantum master equation) but also to the transport rates (that occur in expressions for currents). Focusing on a concrete example, we show that we obtain a compact form for the transient state and for charge- and heat-transport currents. The obtained new "variables" in which the problem can be expressed are invaluable for any detailed analysis of the full time dependence of its solution addressing both the decay rates and the amplitudes in dependence of the initial state and the observables of interest. This detailed analysis combined with numerical calculations is presented elsewhere [19]. Here we focus on the required analytical considerations and calculation of the results. Remarkably, we find for a nontrivial example that without diagonalizing the transition-rate matrix $W$, duality allows the exact time scales of the dynamics of the problem to simply be read off. Moreover, the eigenvectors can be expressed compactly in terms of the decay rates and just a single additional duality-invariant parameter.

Our example is an interacting single-level quantum dot connected to a superconducting contact –a proximized quantum dot– probed by a weakly coupled normal-metal contact, see Fig. 1. So far, applications of fermionic duality mostly focused on models of interacting quantum dots in weak contact with featureless *normal* metals [9, 12, 13], although a possible energy dependence of the coupling to the metal was addressed [14, 16]. When replacing one of the contacts by a superconductor, the open quantum system is modified by the superconducting proximity effect and exhibits a much richer dynamics which is more difficult to analyze and warrants a duality-based approach. The understanding of proximized interacting quantum dots has seen much theoretical progress, see for a review [20]. In particular, in previous work the stationary charge and heat transport were computed [21], as well as the zero- [22] and finite-frequency noise [23] and counting statistics [24]. The more complex driven dynamics of hybrid quantum dots was also analysed and exploited in the context of pumping [25–30]. Furthermore, the dynamics of strongly correlated hybrid quantum dot systems was studied [31–33] but mostly resorted to numerical tools.

Master equations governing proximized interacting quantum dots were microscopically derived using the established real-time approach [34, 35] for the large-gap limit [36, 37] and for a finite gap [21, 38] including coherences in the energy-basis, see also recent work [39]. The time-dependent state of the proximized dot was obtained analytically in Ref. [38], but time-dependent charge and heat currents were not addressed. Yet, we find these two quantities particularly interesting, since they provide experimental access to the complex dynamic behaviour of the proximized quantum dot state, noting also the continued interest in heat transport in superconductors [40–43]. Importantly, for such time-dependent decay the proximity effect requires a careful consideration of the physical procedure of state preparation which enters the master equation as an initial condition. Already in the normal case this warrants a systematic analysis of all possible initial and final gate voltages [9]. For a proximized system the analysis is more complicated since the induced pairing leads to two inequivalent ways of initializing the quantum dot using the gate voltage and requires separate analyses of the distinct ensuing dynamics. This issue has received little attention so far but is essential for the physical predictions as discussed in detail in Ref. [19]. The duality-based solution required for that analysis is presented here and applies independent of the way these initial conditions are realized.

In the present work we show that a duality-based calculation of the time-dependent transport currents provides some important new insights. Importantly, the derivation of fermionic

duality reported in Refs. [9, 16] allows for systems with Hamiltonians with pairing terms describing large gap superconductors. Fermionic duality in this case is a property of the part of the dynamics induced by the *normal* reservoirs and can thus be immediately applied. In doing so we emphasize that the relevance of the different decay rates / time scales of the system evolution [21] is determined by their *amplitudes* and through these also by the initial state. One thus needs to understand the full dependence of the amplitudes on physical parameters. For this purpose, the transition-rate expressions for the master-equation kernel as standardly used [21, 38, 44] are not suitable since they do not take the extremely restrictive fermionic duality into account.

In the limit of vanishing superconducting coupling, we reproduce and extend the results of Ref. [9], which were obtained by exploiting duality for a normal-conducting system. Also, part of our compact expressions recover results of Refs. [21, 38] but in a form more suitable for the decay analysis. Finally, we emphasize that our results apply not only to quantum dots with repulsive interaction ($U > 0$), but also to effectively attractive ones ($U < 0$). Nowadays the latter are also accessible in controlled experiments with a high interest in the interplay with superconductivity [45–48]. Here fermionic duality is particularly relevant since one of its key features is that it inverts the sign of the interaction $U$, thereby connecting the physically very different behaviour of these two classes of systems [14]. We show that only at the crossover between these two cases at $U = 0$ the system has a higher symmetry characterized by a *self-duality* of stationary observables. This ties Coulomb interaction to the breaking of self-duality even in the presence of superconducting pairing.

The paper is organized as follows. After introducing the proximized quantum dot system in Sec. 2, we review the established transport equations in Sec. 3. We present fermionic duality in Sec. 4 and introduce duality invariance which we systematically exploit in Sec. 5 to derive the solution and analyse its general features in Sec. 6. Several limiting cases of the solution are discussed analytically in Sec. 7 and we conclude in Sec. 8. We use units such that $k_B = \hbar = 1$.

## 2 Proximized quantum dot

We consider a quantum dot described by the Hamiltonian $H'_D = \epsilon N + U N_\uparrow N_\downarrow$, with a single level $\epsilon$ controlled by a gate voltage and effective interaction $U$ of either repulsive or attractive sign. Here $N = N_\uparrow + N_\downarrow$ is the electron number operator with $N_\sigma = d_\sigma^\dagger d_\sigma$. The dot is tunnel coupled by $H_T = \sum_{k\sigma} \sqrt{\Gamma/(4\pi)}(d_\sigma^\dagger c_{k\sigma} + \text{h.c.})$ to a metallic reservoir $H_M = \sum_{k\sigma} \omega_k c_{k\sigma}^\dagger c_{k\sigma}$, which is held at temperature $T$ and electrochemical potential $\mu$. The energy-independent dot-metal coupling is assumed weak, $\Gamma \ll T$. The observables of interest are the electron particle current $I_N = \partial_t \langle N_M \rangle$, referred to as charge current for simplicity (the actual charge current being $-|e|I_N$) and the heat current $I_Q = \partial_t \langle H_M - \mu N_M \rangle$, both directed into the metal as in Fig. 1. Here $N_M = \sum_{k\sigma} c_{k\sigma}^\dagger c_{k\sigma}$ is the electron number operator on the metal. The quantum dot is additionally coupled to a superconductor at zero electro-chemical potential $\mu_S = 0$. In the limit of large superconducting gap exceeding all energy scales, the influence of the superconductor on the dot is described [35, 49–52] by a Hamiltonian term with a pairing amplitude $\alpha$:

$$H_S = -\tfrac{1}{2}\alpha d_\uparrow^\dagger d_\downarrow^\dagger + \text{h.c.} \tag{1}$$

We choose the phase $\alpha$ to be *real* by fixing the gauge on the superconductor. It could even be chosen positive but it is convenient not to do so for the duality analysis in Sec. 4. In the resulting Hamiltonian describing the dot-superconductor system, the proximized dot,

$$H_D = H'_D + H_S = \sum_\tau E_\tau |\tau\rangle \langle\tau| + E_1 \sum_\sigma |\sigma\rangle \langle\sigma|, \tag{2}$$

the pairing generates an Andreev splitting $\delta_A \geq |\alpha|$ of the discrete 0- and 2-electron levels [53]. For $\alpha \neq 0$ this splitting differs from the detuning energy $\delta$ as follows:

$$E_\tau = \tfrac{1}{2}(\delta + \tau \delta_A), \quad \begin{cases} \delta &= 2\epsilon + U, \\ \delta_A &= \sqrt{\delta^2 + \alpha^2}. \end{cases} \tag{3}$$

The corresponding states are hybridized into Andreev states with isospin label $\tau = \pm$:

$$|\tau\rangle = \sqrt{\tfrac{1}{2}\left[1 - \tau \frac{\delta}{\delta_A}\right]}|0\rangle - \tau \frac{\alpha}{|\alpha|}\sqrt{\tfrac{1}{2}\left[1 + \tau \frac{\delta}{\delta_A}\right]}|2\rangle . \tag{4}$$

By including $\alpha/|\alpha| = \text{sign}(\alpha)$ into the Andreev states of Ref. [36] we ensure these are eigenstates of the Hamiltonian (2) for all *real* values of $\alpha$ considered later on.[1] The 1-electron spin states $|\sigma\rangle$ of the dot remain unaffected by the pairing and have energy

$$E_1 = \epsilon = \tfrac{1}{2}(\delta - U). \tag{5}$$

Although $\mu$ can be arbitrary relative to the induced *pairing gap* $|\alpha|$ on the dot, the considered model only describes non-equilibrium physics for bias $\mu$ within the *superconductor gap* which is taken infinite here. In this description, quasiparticles of the superconductor are energetically inaccessible. This implies that coherent Cooper pair transfer ($N = 0, 2$ superpositions) within the dot-superconductor system is not internally damped and the only source of dissipation is the weak coupling to the metal probe M. We furthermore focus on the regime considered in Ref. [36] where only the occupation probabilities $\rho_\tau$ of the Andreev states $\tau = \pm$ together with the odd-parity occupation $\rho_1$ need to be considered, i.e. at all times the proximized dot is in a *mixture* of energy states. This is valid provided the induced pairing dominates the dissipation by the metal

$$|\alpha| \gg \Gamma. \tag{6}$$

Of particular interest is the competition of strong Coulomb interaction and strong pairing in the regime $|U|, |\alpha| \gg T \gg \Gamma$ in the full time-dependent transport noting that prior results concern stationary charge [36] and heat-transport [21]. We address this for arbitrary physical parameters and arbitrary initial energy-mixture described independently by a density operator denoted $\rho_0$. In fact, the initial state $\rho_0$ depends on these parameters but in a way that varies with the choice of physical preparation procedure. When analysing and comparing the state and transport dynamics relevant to possible specific experiments this requires separate attention and is considered elsewhere [19].

## 3 Transport equations

### 3.1 Rate and transport equations

Given the parameters and an initial state of the proximized dot $\rho_0$, diagonal in the energy basis, the transport quantities of interest can be obtained from the diagonal elements of the time-evolved state denoted $\rho(t)$. These elements, denoted by $\rho_\pm(t)$ and $\rho_1(t)$, are determined by rate equations derived in Ref. [36] from the microscopic model of Sec. 2:

$$\frac{d\rho_\tau}{dt} = W_{\tau 1}\rho_1 - W_{1\tau}\rho_\tau, \qquad \frac{d\rho_1}{dt} = \sum_\tau W_{1\tau}\rho_\tau - \sum_\tau W_{\tau 1}\rho_1. \tag{7}$$

---

[1]This labelling has the advantage that $\tau = -$ labels the ground state for all *real* $\alpha$, but it implies a swapping $|\tau\rangle \to |-\tau\rangle$ under duality mapping. The latter is in fact desired, see Sec. 4, and is an advantage over the alternative labelling $|\tau'\rangle \equiv |\tau \alpha/|\alpha|\rangle$, which leaves the states invariant.

Here $\rho_\tau$ is the occupation probability of the even-parity state $|\tau\rangle$ with $\tau = \pm$ and $\rho_1$ is the occupation – summed over spin $\sigma = \uparrow, \downarrow$ – of the odd-parity states. The formula for the charge current into the metal derived in Ref. [36] is

$$I_N = \sum_{\eta\tau} \eta \left[ W_{1,\tau}^\eta \rho_\tau + W_{\tau,1}^\eta \rho_1 \right], \tag{8}$$

where $\eta = \pm$ counts the number of electrons transferred to the metal (in units of $-e$). Counting instead the energy transferred, gives a similar formula for the energy current

$$I_E = -\sum_{\eta\tau} \left[ (E_1 - E_\tau) W_{1,\tau}^\eta \rho_\tau + (E_\tau - E_1) W_{\tau,1}^\eta \rho_1 \right], \tag{9}$$

from which the heat current $I_Q = I_E - \mu I_N$ into the metal follows. The transition rates in the master equation (7) read

$$W_{1,\tau} = \sum_\eta W_{1,\tau}^\eta, \qquad W_{\tau,1} = \sum_\eta W_{\tau,1}^\eta, \tag{10}$$

whereas the transport rates featuring in the current formulas (8)-(9),

$$W_{1,\tau}^\eta = \Gamma_{\eta\tau} f^{-\eta}(E_{\eta,\tau} - \mu), \qquad W_{\tau,1}^\eta = \tfrac{1}{2}\Gamma_{\bar{\eta}\tau} f^{+\bar{\eta}}(E_{\bar{\eta},\tau} - \mu), \tag{11}$$

explicitly keep track of whether an electron, is transferred to ($\eta = -$) or from ($\eta = +$) the proximized dot with probability $f^{-\eta}(\omega) = (e^{-\eta\omega/T} + 1)^{-1}$. Importantly, spin degrees of freedom have already been eliminated being incorporated into the factor $1/2$ distinguishing the rates (11). Thus, $\eta$ indicates the direction relative to the positive particle current into the metal as in Fig. 1 and we denote the opposite by $\bar{\eta} \equiv -\eta$. The addition energies are denoted by[2]

$$E_{\eta,\tau} = \eta\left(E_\tau - E_1\right) = \tfrac{1}{2}\left(\eta\tau\delta_{\text{A}} + \eta U\right), \tag{12}$$

and the effective rates leading to these transitions on the proximized dot are proportional to the probabilities of $|0\rangle$ and $|2\rangle$ in the superposition $|\tau\rangle$ [Eq. (4)]:

$$\Gamma_{\eta\tau} = \gamma_p \tfrac{1}{2}\left(1 + \eta\tau\frac{\delta}{\delta_{\text{A}}}\right). \tag{13}$$

Importantly, $\gamma_p = \Gamma$ denotes the lump sum of the tunnel rates that arises when eliminating spin. The advantage of the formal analysis presented next lies in avoiding the consideration of these individual processes $(\eta, \tau)$ and in working with (anti-)symmetric combinations of transport rates dictated by fermionic duality [Eq. (49)], instead.

## 3.2 Liouville-space formulation

Although equations (7)-(8) were derived in Ref. [36] the time-dependent solutions for transient transport quantities (8)-(9) of interest here were not given. To obtain the simplest form of these solutions by exploiting symmetries, it is crucial to start from equations in basis-independent form in contrast to Eqs. (7)-(9) where the basis was fixed. Using Liouville-space notation, any operator $x$ is written as a supervector $|x) = x$ and its adjoint, the left super-covector $(x| = \text{tr}(x^\dagger \bullet)$ denotes a function on operators such that $(y|x) = \text{tr}(y^\dagger x)$. The rate

---

[2] Our notation merely relabels Eq. (2.6) of Ref. [36], $E_{\eta,\tau}^{[36]} \equiv \eta(E_{\eta\tau} - E_1) = E_{\eta,(\eta\tau)}$ by setting $\tau \to \eta\tau$.

equations refer to a basis of pure Andreev states for the parity $+1$ sector ($N = 0, 2$) and a uniform spin-mixed state for the parity $-1$ sector ($N = 1$), respectively,

$$|\tau) \equiv |\tau\rangle\langle\tau|, \quad \tau = \pm, \qquad |1) \equiv \tfrac{1}{2}\sum_\sigma |\sigma\rangle\langle\sigma|. \tag{14}$$

Note that $(\tau|\tau) = 1$ but $(1|1) = \frac{1}{2}$ is not normalized due to the elimination of spin. As a result, the completeness relation for the identity superoperator reads

$$\mathcal{I} = \sum_\tau |\tau)(\tau| + 2|1)(1|. \tag{15}$$

Applied to a state, one obtains its expansion in terms of probabilities $\rho_\lambda \equiv (\lambda|\rho)/(\lambda|\lambda)$

$$|\rho) = \sum_{\tau=\pm} \rho_\tau|\tau) + \rho_1|1). \tag{16}$$

The rate equations (7) can now be written as a basis-independent master equation

$$\frac{d}{dt}|\rho(t)) = W|\rho(t)), \qquad |\rho(0)) = |\rho_0), \tag{17}$$

for the state supervector when using the rate superoperator

$$W = -\sum_\tau W_{1,\tau}|\tau)(\tau| + \sum_\tau W_{1,\tau}|1)(\tau| + \sum_\tau 2W_{\tau,1}|\tau)(1| - \sum_\tau 2W_{\tau,1}|1)(1|, \tag{18}$$

as one verifies by inserting $\mathcal{I}$ into Eq. (17) and using Eq. (14). To solve the evolution problem we have to diagonalize $W$ and explicitly express the formal solution

$$|\rho(t)) = e^{Wt}|\rho_0), \tag{19}$$

in the *right* eigenvectors of $W$. The final step is to compute the amplitudes as functions of the initial state $|\rho_0)$. Similarly, the charge current (8) and energy current (9) can be expressed as scalar products with two basis-independent left supervectors:

$$I_N(t) = (I_N|\rho(t)), \qquad (I_N| = \sum_{\eta\tau} \eta\Big[W^\eta_{1,\tau}(\tau| + 2W^\eta_{\tau,1}(1|\Big], \tag{20a}$$

$$I_E(t) = (I_E|\rho(t)), \qquad (I_E| = -\sum_{\eta\tau} E_{+,\tau}\Big[-W^\eta_{1,\tau}(\tau| + 2W^\eta_{\tau,1}(1|\Big]. \tag{20b}$$

To solve the transport problem, we need to express these in the *left* eigenvectors of $W$, compute their coefficients and evaluate the scalar products with the evolving state (19).

## 4 Fermionic duality

To diagonalize the rate superoperator $W$ and analyse the non-trivial parameter dependence of the resulting solution, the above form (18) based on *energy basis* matrix elements of $W$ ($\propto W_{1\tau}$ or $W_{\tau 1}$) is not a good starting point (even though it is suitable for the microscopic derivation of $W$ [36]). It is important to see why this is the case: for a large class of models including the present system the rate superoperator of the master equation obeys a fermionic duality relation [9, 10, 15]:

$$W + \tfrac{1}{2}\gamma_p \mathcal{I} = -\Big[\mathcal{P}\overline{\big(W + \tfrac{1}{2}\gamma_p \mathcal{I}\big)}\mathcal{P}\Big]^\dagger. \tag{21}$$

This symmetry is truly dissipative since it explicitly features the lump sum rate $\gamma_p$ [Eq. (13)]. As written here, it states that the *shifted* rate superoperator is *invariant* under a duality mapping up to a sign. The duality mapping is a "generalized hermitian adjoint" consisting of three parts: the ordinary hermitian adjoint †, the superoperator $\mathcal{P} = p \bullet$ which left-multiplies its argument $\bullet$ with the fermion-parity operator $p = (-\mathbb{1})^N$ and the over-bar which denotes an inversion of parameters. For our model the latter reads

$$\overline{X(\epsilon, U, \alpha, \mu)} = X(\bar{\epsilon}, \bar{U}, \bar{\alpha}, \bar{\mu}), \qquad (22)$$

where $X$ denotes some function and $\bar{x} = -x$ for $x = \epsilon, U, \alpha, \mu$. Temperature $T$ and coupling $\Gamma$ are left untouched by this inversion and were therefore not indicated in Eq. (22). In the weak coupling limit considered here, inversion of $\Gamma$ in the general duality result [9, 10] mentioned in the introduction, can be usefully avoided by convention [9, 16].[3]

Unlike the earlier applications of duality involving only normal reservoirs, we here have to deal with *parameter-dependent* bases due to the superconducting pairing $\alpha$ introduced via the dot Hamiltonian. For example, for the energy eigenbasis [Eqs. (4),(14)]:

$$\left[\mathcal{P}\overline{|\tau)}\right]^{\dagger} = (-\tau|, \quad \left[\mathcal{P}\overline{|1)}\right]^{\dagger} = -(1|, \qquad (23)$$

since $\mathcal{P}|1) = -|1)$ and $\mathcal{P}|\tau) = |\tau)$ and $\overline{|1)} = |1)$ while the energy eigenstates created by pairing are swapped $\overline{|\pm)} = |\mp)$, when inverting the signs of $\delta$ and $\alpha$ in Eq. (4). Fermionic duality makes explicit why working in this basis is neither suitable for finding the solution nor for performing the analysis of its properties: From Eq. (21) we see that the rate superoperator $W$ shifted by $\gamma_p/2$ is antisymmetric with respect to the three-fold duality mapping even though $W$ itself is not (anti-)hermitian $W \neq \pm W^{\dagger}$. This relation immediately implies that left and right eigenvectors of $W + \frac{1}{2}\gamma_p \mathcal{I}$ –and thus of $W$– for in general *different* eigenvalues labelled by their operators $x, y$ are *cross*-related in pairs

$$\left[\mathcal{P}\overline{|x)}\right]^{\dagger} \propto (y|, \qquad \left[\overline{(x|\mathcal{P}}\right]^{\dagger} \propto |y). \qquad (24)$$

Both are equivalent to the operator equation $y = p\bar{x}$. The corresponding eigenvalues obey

$$\lambda_x = -\bar{\lambda}_y. \qquad (25)$$

In order to take advantage of these constraints imposed by the fermionic duality one should work in an orthogonal basis (i) which has the same property (24) as the desired eigenvectors, for which (ii) the matrix elements of $W$ transform in the same way as the desired eigenvalues under the scalar duality mapping, the overbar (22). While the energy basis does have property (i) by Eq. (23), it fails to have (ii). In particular, the basis-independent duality (21) in the energy basis implies cross-relations between off-diagonal matrix elements which involves parameter inversion (22), such as $W^{\eta}_{1\tau} \propto \bar{W}^{\eta}_{\bar{\tau}1}$ [Eq. (26)]. Instead, in a duality-adapted basis [Eq. (36)] matrix elements of $W$ are not only cross-related, but even invariant under the map (22).

## 5 Solution using fermionic duality

Thus guided by duality, we now first identify rate variables which are *invariant* under the mapping Eq. (22) up to a sign and then identify cross-related basis supervectors for which

---

[3]Temperature, however, plays a fundamentally different role in fermionic duality, in particular, the $T \to \infty$ limit, and is not touched. See Ref. [9, 16] for more details.

matrix elements of $W$ are proportional to these variables. Although not smaller in number, these variables significantly simplify the derivation and analysis of the solution as compared to a brute-force linear algebra approach. We also stress that the brute-force approach cannot achieve such simplifications since fermionic duality exploits the *functional parameter dependence* of rate expressions and not just their mutual linear dependence.

## 5.1 Duality-invariant rate variables

As illustrated in Ref. [9] for a simple, weak-coupling example, fermionic duality is based on the behaviour of the fermionic reservoir distribution function $f^\eta(\omega) = (e^{\eta\omega} + 1)^{-1} = f^{-\eta}(-\omega)$ under inversion of the energy: $f^\eta(\omega) = 1 - f^\eta(-\omega)$. It implies relations between actual and dual transport rates (11):

$$W_{1,\tau}^\eta = \Gamma_{\eta\tau} - \bar{W}_{1,\bar\tau}^\eta, \qquad 2W_{\tau,1}^\eta = \Gamma_{\bar\eta\tau} - 2\bar{W}_{\bar\tau,1}^\eta, \qquad W_{1,\tau}^\eta = 2\bar{W}_{\bar\tau,1}^{\bar\eta}. \tag{26}$$

In the last relation, inverting the direction of the transition from $\tau \to 1$ to $1 \to \tau$ inverts both the state ($\bar\tau = -\tau$) and the direction of electron transfer ($\bar\eta = -\eta$). Note how our convention for inverting indices [Eq. (11) ff.] naturally combines with the notation for parameter inversion of functions [Eq. (22)]. These relations allow to eliminate transport / transition rates in one direction in favour of those for the opposite direction by a sum rule obtained by combining the second and third relation of (26):

$$2W_{\tau,1}^\eta + W_{1,\tau}^{\bar\eta} = \Gamma_{\bar\eta\tau}, \qquad 2W_{\tau,1} + W_{1,\tau} = \gamma_p, \tag{27}$$

where the second expression results from summing over $\eta$. We can thus focus on parametrizing the rates for one direction, say $\tau \to 1$. Using relations (26) duality-invariant variables can be obtained as follows by considering (anti-) symmetry with respect to inversion of the energy-state ($\tau$) or electron-transfer direction ($\eta$) or both. We first decompose the right hand side of the first sum-rule (27) into even and odd parts, using Eq. (13) with $\eta, \tau = \pm$,

$$\Gamma_{\eta\tau} = \tfrac{1}{2}\big(\gamma_p + \eta\tau\gamma_p'\big), \qquad \gamma_p \equiv \Gamma, \qquad \gamma_p' \equiv \gamma_p \frac{\delta}{\delta_A}. \tag{28}$$

Here $\gamma_p$ is the simple lump sum rate of Eq. (13) featuring in the duality relation (21). The coefficient $\gamma_p'$ is also relatively simple, adding only a dependence on the detuning relative to the pairing, $\gamma_p'/\gamma_p = \delta/\delta_A = (\delta/|\delta|)/\sqrt{1 + (\alpha/\delta)^2}$. Next we decompose the transport rates by making an ansatz for $W_{1,\tau}^\eta$ and then use the first sum rule (27) for $W_{\tau,1}^\eta$

$$W_{1,\tau}^\eta = \tfrac{1}{2}\Gamma_{\eta\tau} + \tfrac{1}{2}\big[\gamma_C + \eta\gamma_c' + \tau\gamma_s + \eta\tau\gamma_S'\big], \tag{29a}$$

$$2W_{\tau,1}^\eta = \tfrac{1}{2}\Gamma_{\bar\eta\tau} - \tfrac{1}{2}\big[\gamma_C - \eta\gamma_c' + \tau\gamma_s - \eta\tau\gamma_S'\big]. \tag{29b}$$

Inserting the first ansatz into the first of Eqs. (26) and summing over $\eta, \tau = \pm$ as either $\sum_{\eta\tau}$ or $\sum_{\eta\tau} \eta$ or $\sum_{\eta\tau} \tau$ or $\sum_{\eta\tau} \eta\tau$, we see that indeed the new variables are "($\pm$) invariant" under the scalar duality mapping of physical parameters:

$$\gamma_p = +\bar\gamma_p, \qquad\qquad \gamma_C = -\bar\gamma_C, \qquad\qquad \gamma_s = +\bar\gamma_s, \tag{30a}$$

$$\gamma_p' = -\bar\gamma_p', \qquad\qquad \gamma_c' = -\bar\gamma_c', \qquad\qquad \gamma_S' = +\bar\gamma_S'. \tag{30b}$$

Therefore these are the appropriate variables in which we should express the shifted rate superoperator $W + (\gamma_p/2)\mathcal{I}$ from the very beginning to optimally exploit the duality relation (21). The prime notation is physically motivated: Unprimed (= $\eta$-symmetric) invariants determine the *state* evolution (insensitive to electron transfer direction) whereas the primed (= $\eta$-antisymmetric) invariants enter only into *transport* quantities (sensitive to electron-transfer

direction, cf. Eq. (64b)). The motivation of subscripts $p$ (parity), $C$ (charge) and $S$ (isospin polarization)[4] will become clear below.

Thus, using only duality relations (26), we can already conclude that the most non-trivial parameter dependence of the solution of the time-dependent problem must be contained in the 4 invariants $\gamma_C$, $\gamma_c'$, $\gamma_s$, $\gamma_S'$. They can be explicitly expressed as linear combinations of transport rates (anti)symmetrized with respect to $\eta$ and / or $\tau$ [Eq. (45),(49),(74)] but these should be avoided until *after* we have solved the problem posed in Sec. 6.2 and turn to analysis of the parameter dependence.

## 5.2 Duality-adapted basis

Having exploited the duality for the scalar rate coefficients we should also make use of it for the choice of supervector basis. To this end, we use that in addition to the mapping of scalars to a dual model (indicated by the overbar ‾), the duality additionally involves the mappings $|\bullet)^{\dagger} = (\bullet|$ and $\mathcal{P}|\bullet) = |p\,\bullet)$ involving super(co)vectors. To decompose the shifted rate superoperator $W + \frac{1}{2}\gamma_p \mathcal{I}$ whose support is spanned by the energy-diagonal supervectors $\{|+), |-), |1)\}$ we introduce a new basis of right vectors

$$|1) = \sum_{\tau} |\tau) + 2|1), \qquad |p) = \sum_{\tau} |\tau) - 2|1), \qquad |A) \equiv \sum_{\tau} \tau|\tau), \qquad (31)$$

and corresponding adjoint left vectors $(1|$, $(A|$, and $(p|$ orthogonal to these with respect to the scalar product $(y|x) = \text{tr}(y^{\dagger}x)$. Here $1$ is the identity operator, $p$ is the parity operator [featuring in the duality relation (21)] and $A$ is the polarization operator of the Andreev states (superconducting Bloch-vector or isospin, see Eq. (56) later).

Although other choices are possible, this basis follows right away from duality noting only that any rate superoperator has the identity as a left zero eigenvector by probability normalization, $(1|W = 0$. This implies that its *duality transform*, $[\overline{(1|\mathcal{P}]^{\dagger}} = |p)$, is a right eigenvector of $W$. Thus we should include both $|1)$ and $|p)$ in the orthogonal basis, fixing the remaining basis vector to be $|A)$ when normalized as follows

$$\tfrac{1}{4}(1|1) = \tfrac{1}{4}(p|p) = \tfrac{1}{2}(A|A) = 1\,. \qquad (32)$$

The identity superoperator projecting on the support of the energy basis is thus

$$\mathcal{I} = \tfrac{1}{4}|1)(1| + \tfrac{1}{2}|A)(A| + \tfrac{1}{4}|p)(p|\,. \qquad (33)$$

Importantly –like the sets of biorthogonal left and right eigenvectors of $W$ that we seek [Eq. (24)]– the left and right orthogonal basis vectors are *cross*-related by the duality mapping: using the operator relations $pA = A$ and $p^2 = 1$

$$\overline{\mathcal{P}|1)}^{\dagger} = (p|, \quad \overline{\mathcal{P}|A)}^{\dagger} = -(A|, \quad \overline{\mathcal{P}|p)}^{\dagger} = (1|, \qquad (34a)$$

$$\overline{(1|\mathcal{P}}^{\dagger} = |p), \quad \overline{(A|\mathcal{P}}^{\dagger} = -|A), \quad \overline{(p|\mathcal{P}}^{\dagger} = |1)\,. \qquad (34b)$$

The *parameter-dependent* observable $A$ has the scalar duality mapping $\bar{A} = -A$ [ Eq. (31) and (23)]. For later reference we note the inverse formulas of Eq. (31):

$$|\tau) = \tfrac{1}{4}\big[|1) + |p)\big] + \tau\tfrac{1}{2}|A), \quad \tau = \pm, \qquad |1) = \tfrac{1}{4}\big[|1) - |p)\big]\,. \qquad (35)$$

By merely changing to the duality-adapted variables (29) and basis (35) the form of the rate superoperator (18) shifted by the identity (33) to $W + (\gamma_p/2)\mathcal{I}$ is greatly simplified:

$$W + \tfrac{1}{2}\gamma_p \mathcal{I} = \tfrac{1}{2}\gamma_p \tfrac{1}{4}\big[|1)(1| - |p)(p|\big] - \gamma_C \tfrac{1}{2}\big[|p)(1| + |A)(A|\big] - \gamma_s \tfrac{1}{2}\big[|p)(A| + |A)(1|\big]\,. \qquad (36)$$

---

[4]None of the duality invariants is related to the real spin which was eliminated already in Eq. (7)-(9).

This form manifests fermionic duality (21) term-by-term: the first and last superoperators are $(-)$ invariant [by Eq. (34)], while their coefficients are $(+)$ invariant [Eq. (46)] and vice versa for the middle term. Thus $W + \frac{1}{2}\gamma_p \mathcal{I}$ is a $(-)$ invariant explicitly confirming Eq. (21). Therefore this is the appropriate basis in which to expand $W + (\gamma_p/2)\mathcal{I}$ in order to optimally exploit the duality (21) for its diagonalization.

### 5.3 Diagonalization of the rate superoperator

By changing to the duality-adapted basis we have eliminated 3 out of 9 matrix elements and obtained a lower-triangular form with only 3 independent duality-invariant matrix elements. This simplifies the calculation of the diagonal form in several ways:

$$W + \tfrac{1}{2}\gamma_p \mathcal{I} = \tfrac{1}{2}\gamma_p |z)(z'| - \gamma_C |c)(c'| - \tfrac{1}{2}\gamma_p |p)(p'|. \tag{37}$$

(i) The eigenvalues can be read off from the diagonal of Eq. (36) giving $\pm\frac{1}{2}\gamma_p$ and $-\gamma_C$ noting the normalization (33). (ii) The first left basis vector $(\mathbb{1}|$ and the last right basis vector $|p)$ are eigenvectors for different eigenvalues. The two remaining left (right) eigenvectors are then found by standard forward (backward) recursion for lower-triangular matrices:

$$(z'| = (\mathbb{1}|, \tag{38a}$$

$$(c'| = (A| + \frac{\gamma_s}{\frac{1}{2}\gamma_p + \gamma_C}(\mathbb{1}|, \tag{38b}$$

$$(p'| = \tfrac{1}{4}(p| + \frac{\gamma_s}{\frac{1}{2}\gamma_p - \gamma_C}\tfrac{1}{2}(A| + \Big(\frac{\gamma_C}{\gamma_p} + \frac{\gamma_s}{\gamma_p}\frac{\gamma_s}{\frac{1}{2}\gamma_p - \gamma_C}\Big)\tfrac{1}{2}(\mathbb{1}|, \tag{38c}$$

$$|z) = \tfrac{1}{4}|\mathbb{1}) - \frac{\gamma_s}{\frac{1}{2}\gamma_p + \gamma_C}\tfrac{1}{2}|A) - \Big(\frac{\gamma_C}{\gamma_p} - \frac{\gamma_s}{\gamma_p}\frac{\gamma_s}{\frac{1}{2}\gamma_p + \gamma_C}\Big)\tfrac{1}{2}|p), \tag{38d}$$

$$|c) = \tfrac{1}{2}\left[|A) - \frac{\gamma_s}{\frac{1}{2}\gamma_p - \gamma_C}|p)\right], \tag{38e}$$

$$|p) = |(-\mathbb{1})^N). \tag{38f}$$

This compact result reveals that the coefficients of the different eigenvectors in fact have a very similar functional form. Notably, the eigen*vectors* are specified by the eigen*values*, namely the duality invariants $\pm\frac{1}{2}\gamma_p$ and $\gamma_C$, and *a single* additional parameter, the duality invariant $\gamma_s$. (iii) Due to their recursive construction, the normalization of the eigenvectors is automatically fixed by that of the basis vectors:

$$(z'|z) = \tfrac{1}{4}(\mathbb{1}|\mathbb{1}) = 1, \qquad (c'|c) = \tfrac{1}{2}(A|A) = 1, \qquad (p'|p) = \tfrac{1}{4}(p|p) = 1. \tag{39}$$

(iv) Finally, the cross-relation of eigenvectors (24) dictated by duality is manifest,

$$\overline{\mathcal{P}|z)}^\dagger = (p'|, \qquad \overline{\mathcal{P}|c)}^\dagger = -\tfrac{1}{2}(c'|, \qquad \overline{\mathcal{P}|p)}^\dagger = (z'|, \tag{40a}$$

$$\overline{(z'|\mathcal{P}}^\dagger = |p), \qquad \overline{(c'|\mathcal{P}}^\dagger = -2|c), \qquad \overline{(p'|\mathcal{P}}^\dagger = |z), \tag{40b}$$

by merely noting the cross-relation of the duality-adapted basis vectors [Eq. (34)] and the duality invariance of the scalar coefficients [Eq. (46)]. This mapping preserves the bi-orthogonality of the left and right eigenvectors. It also shows how the spectral form (37) of $W + \frac{1}{2}\gamma_p \mathcal{I}$ ensures duality $(-)$ invariance: The mapping cross relates the first and last spectral projectors and it relates the middle spectral projector to itself.[5] Together with the $(+)$ invariance of the

---

[5]The left and right eigenvectors to middle eigenvalue $-\gamma_C$ cannot be normalized to eliminate the factors $-1/2$ resp. $-2$ without undesirable effects. The $-$ sign *can* be eliminated at the expense of normalization factors which are either complex (making $c$, $c'$ non-self adjoint) or discontinuous in $\delta$, both unnecessary complications. The factor $1/2$ *can* be eliminated by normalization $1/\sqrt{2}$ but our choice ensures that $(c'|$ occurs without pre-factors in the charge current operators (which later on cancel in expectation values).

eigenvalue $\pm\frac{1}{2}\gamma_p$ and the $(-)$ invariance of the eigenvalue $\gamma_C$ this implies that Eq. (37) simply inverts its sign as it should by Eq. (21). Note carefully that this inversion is not simply achieved by inverting the signs of all eigenvalues.

## 5.4 Stationary observables and their duals

We now see that we can entirely express the eigenvectors (38) of $W$ in terms of *expectation values* of *physical observables* $\mathbb{1}$, $A$ and $p$ with respect to *two* stationary states, that of the actual system, $|z)$, and of the *dual* system with inverted parameters, $|\bar{z})$ [Eq. (22)]:

$$(z'| = (\mathbb{1}|, \qquad |z) = \tfrac{1}{4}|\mathbb{1}) + \langle A\rangle_z \tfrac{1}{2}|A) + \langle p\rangle_z \tfrac{1}{4}|p), \tag{41a}$$

$$(c'| = (A| - \langle A\rangle_z (\mathbb{1}|, \qquad |c) = \tfrac{1}{2}\big[|A) - \langle A\rangle_{\bar{z}}|p)\big], \tag{41b}$$

$$(p'| = \tfrac{1}{4}(p| + \langle A\rangle_{\bar{z}} \tfrac{1}{2}(A| + \langle p\rangle_{\bar{z}} \tfrac{1}{4}(\mathbb{1}|, \qquad |p) = |(-\mathbb{1})^N). \tag{41c}$$

Here $\langle\bullet\rangle_z = (\bullet|z)$ takes the stationary average and $\langle\bullet\rangle = (\bullet|\bar{z})$ takes the average for the stationary state of the dual system,

$$|\bar{z}) = \tfrac{1}{4}|\mathbb{1}) + \langle A\rangle_{\bar{z}} \tfrac{1}{2}|A) + \langle p\rangle_z \tfrac{1}{4}|p). \tag{42}$$

In the dual of Eq. (41a), $\bar{A} = -A$ occurs twice such that the signs cancel. One verifies that applying $(\mathbb{1}|$, $(A|$, $(p|$ to Eq. (41a) indeed gives the expectation values of these observables as written. These averages can be expressed in terms of only three invariants

$$\langle A\rangle_z = -\frac{\gamma_s}{\tfrac{1}{2}\gamma_p + \gamma_C}, \quad \langle p\rangle_z = -\frac{2\gamma_C}{\gamma_p} + \frac{2\gamma_s^2}{\gamma_p(\tfrac{1}{2}\gamma_p + \gamma_C)}, \tag{43a}$$

$$\langle A\rangle_{\bar{z}} = \frac{\gamma_s}{\tfrac{1}{2}\gamma_p - \gamma_C}, \quad \langle p\rangle_{\bar{z}} = \frac{2\gamma_C}{\gamma_p} + \frac{2\gamma_s^2}{\gamma_p(\tfrac{1}{2}\gamma_p - \gamma_C)}. \tag{43b}$$

Clearly, in each column of Eq. (43) the dual expressions transform into each other by $\gamma_C \to -\gamma_C$ and $\gamma_s \to \gamma_s$ [Eq. (30)], noting that $p \to \bar{p} = p$ but $A \to \bar{A} = -A$ [Eq. (34)]. The mere form of Eqs. (41) already guarantees the bi-orthogonality of the eigenvectors with one important exception: the biorthogonality

$$(p'|z) = \tfrac{1}{4}\big[\langle p\rangle_{\bar{z}} + \langle p\rangle_z\big] + \tfrac{1}{2}\langle A\rangle_z\langle A\rangle_{\bar{z}} = 0, \tag{44}$$

expresses a non-trivial relation between pairs of stationary expectation values of the physical system and of the dual system. This is verified to hold by inserting Eqs. (43), but was overlooked in earlier related work [9], see Sec. 7.1. That such an additional constraint exists could however be expected since we have four quantities depending on only three invariants $\gamma_p$, $\gamma_C$ and $\gamma_s$.

Having fully exploited the duality for diagonalizing $W + \tfrac{1}{2}\gamma_p\mathcal{I}$, we can now return to the rate superoperator $W$ of interest by shifting back, introducing

$$\gamma_c \equiv \gamma_C + \tfrac{1}{2}\gamma_p, \qquad \gamma_s' \equiv \gamma_S' + \tfrac{1}{2}\gamma_p'. \tag{45}$$

Although these expressions are no longer strictly invariant [Eq. (30)], they still obey *shifted* invariance [Eq. (28)]:

$$\gamma_c = \gamma_p - \bar{\gamma}_c, \qquad \gamma_s' = \bar{\gamma}_s' - \bar{\gamma}_p'. \tag{46}$$

Inserting Eq. (36) and (37) into $W = (W + \tfrac{1}{2}\gamma_p\mathcal{I}) - \tfrac{1}{2}\gamma_p\mathcal{I}$ using Eq. (33) we obtain

$$W = \gamma_p \tfrac{1}{4}\big[|p)(\mathbb{1}| - |p)(p|\big] - \gamma_c \tfrac{1}{2}\big[|p)(\mathbb{1}| + |A)(A|\big] - \gamma_s \tfrac{1}{2}\big[|p)(A| + |A)(\mathbb{1}|\big] \tag{47}$$

$$= -\gamma_c|c)(c'| - \gamma_p|p)(p'|. \tag{48}$$

Although the first line also has lower-triangular form as in Eq. (36) with even an extra zero element, the duality relation (21) has become less clear by un-doing the shift.[6] Nevertheless, we can now directly work with $W$ instead of its shifted version and the appropriate variables are then $\gamma_p, \gamma_p', \gamma_c, \gamma_c', \gamma_s, \gamma_s'$. These still have the nice feature that they are linear combinations of transport rates (anti)symmetrized with respect to $\eta$ and/or $\tau$:

$$\gamma_p \equiv \tfrac{1}{2}\sum_{\eta\tau}\left(W_{1,\tau}^{\eta} + 2W_{\tau,1}^{\bar\eta}\right), \qquad \gamma_c \equiv \tfrac{1}{2}\sum_{\eta\tau}W_{1,\tau}^{\eta}, \qquad \gamma_s \equiv \tfrac{1}{2}\sum_{\eta\tau}\tau W_{1,\tau}^{\eta}, \tag{49a}$$

$$\gamma_p' \equiv \tfrac{1}{2}\sum_{\eta\tau}\eta\tau\left(W_{1,\tau}^{\eta} + 2W_{\tau,1}^{\bar\eta}\right), \qquad \gamma_c' \equiv \tfrac{1}{2}\sum_{\eta\tau}\eta W_{1,\tau}^{\eta}, \qquad \gamma_s' \equiv \tfrac{1}{2}\sum_{\eta\tau}\eta\tau W_{1,\tau}^{\eta}. \tag{49b}$$

The last two columns follow from Eqs. (28)-(29) by inserting the shifted invariants (45),

$$W_{1,\tau}^{\eta} = \tfrac{1}{2}\left(\gamma_c + \eta\gamma_c' + \tau\gamma_s + \eta\tau\gamma_s'\right), \tag{50a}$$

$$2W_{\tau,1}^{\eta} = \tfrac{1}{2}\left((\gamma_p - \gamma_c) + \eta\gamma_c' - \tau\gamma_s + \eta\tau(-\gamma_p' + \gamma_s')\right), \tag{50b}$$

and then summing either as $\sum_{\eta\tau}$ or $\sum_{\eta\tau}\eta$ or $\sum_{\eta\tau}\tau$ or $\sum_{\eta\tau}\eta\tau$. With respect to the (shifted) invariants, $\gamma_p, \gamma_s, \gamma_c$, the expectation values of operators of Eq. (43) read

$$\langle A\rangle_z = -\frac{\gamma_s}{\gamma_c}, \qquad \langle p\rangle_z = 1 - 2\frac{\gamma_c^2 - \gamma_s^2}{\gamma_p\gamma_c}, \tag{51a}$$

$$\langle A\rangle_{\bar z} = \frac{\gamma_s}{\gamma_p - \gamma_c}, \qquad \langle p\rangle_{\bar z} = 1 - 2\frac{(\gamma_p - \gamma_c)^2 - \gamma_s^2}{\gamma_p(\gamma_p - \gamma_c)}. \tag{51b}$$

## 5.5 Evolving state supervector and duality-invariant physical constraints

We can now express the evolving state supervector in the eigenbasis of the evolution

$$|\rho(t)) = e^{Wt}|\rho_0) = |z) + |c)e^{-\gamma_c t}(c'|\rho_0) + |p)e^{-\gamma_p t}(p'|\rho_0), \tag{52}$$

with amplitudes depending on physical expectation values in the initial state ($\rho_0$), the stationary state ($z$) *and* the dual stationary state ($\bar z$):

$$(z'|\rho_0) = 1, \tag{53a}$$

$$(c'|\rho_0) = \langle A\rangle_{\rho_0} - \langle A\rangle_z, \tag{53b}$$

$$(p'|\rho_0) = \tfrac{1}{4}\left[\langle p\rangle_{\rho_0} + \langle p\rangle_{\bar z}\right] + \tfrac{1}{2}\langle A\rangle_{\bar z}\langle A\rangle_{\rho_0} \tag{53c}$$

$$= \tfrac{1}{4}\left[\langle p\rangle_{\rho_0} - \langle p\rangle_z\right] + \tfrac{1}{2}\langle A\rangle_{\bar z}\left[\langle A\rangle_{\rho_0} - \langle A\rangle_z\right], \tag{53d}$$

subtracting relation (44) from Eq. (53c) in the last step. At this point it is relevant to consider the physical constraints on the duality invariants and the stationary values of duality-adapted observables which will be used later on.

First, the decay rates of the time-dependent state (52), the (shifted) duality invariants $\gamma_c$ and $\gamma_p$, are clearly non-negative since they are proportional to the sums (49a) of non-negative transition rates of the master equation. However, their individual non-negativity imposes stronger physical constraints which also involve the invariant $\gamma_s$ which can be negative like $\gamma_c'$

---

[6]In the cross relation of spectral projectors the physically important zero eigenvalue projector is only implicitly defined by its bi-orthogonality to the non-zero eigenvalue projectors. The zero eigenvalue thus remains implicit until it appears in the cross-relation of the spectrum, mapping $(0, \gamma_c, \gamma_p)$ to $(\gamma_p, \gamma_p - \gamma_c, 0)$.

and $\gamma_s'$. These are not rates but "rate asymmetries" Eq. (49b) which account for various *differences* of the transport rates [electron/hole ($\eta$) and state asymmetry ($\tau$)]. The non-negativity of $W_{1,\tau}$ and $W_{\tau,1}$ [Eq. (11)] imposes a bound on the negativity of $\gamma_s$:

$$\gamma_p \geq 0, \quad \gamma_c, \gamma_p - \gamma_c \geq |\gamma_s|. \tag{54}$$

Thus the decay rate $\gamma_c$ should not only be non-negative but even larger than the magnitude of the rate-asymmetry $\gamma_s$. Likewise, the decay rate $\gamma_p$ should not only exceed the decay rate $\gamma_c$, but it must do so by more than the magnitude of $\gamma_s$. The latter two conditions are equivalent to one, duality-invariant condition on $\gamma_C = \gamma_c - \gamma_p/2$:

$$|\gamma_C| = |\gamma_c - \tfrac{1}{2}\gamma_p| \leq \left||\gamma_s| - \tfrac{1}{2}\gamma_p\right|, \tag{55}$$

expressing that the sum of transition rates $2\gamma_c = \sum_\tau W_{1,\tau}$ is always closer to $\gamma_p$ than their difference $2\gamma_s = \sum_\tau \tau W_{1,\tau}$. Interestingly, whether $\gamma_c$ dominates $\gamma_p/2$ or vice versa is determined by the sign of $\gamma_C$ which can be either $+$ or $-$ [see Eq. (63) below].

Using this we can explicitly see that the stationary supervector (41a) giving the long-time limit of (52) is a valid physical state for all parameter values of the model by rewriting it as a sum of two operators with parity $+1$ and $-1$:

$$|z) = \left\{\tfrac{1}{2}\left[1 + \langle p\rangle_z\right]\tfrac{1}{4}\left[|\mathbb{1}) + |p)\right] + \tfrac{1}{2}\langle A\rangle_z|A)\right\} + \tfrac{1}{2}\left[1 - \langle p\rangle_z\right]\tfrac{1}{4}\left[|\mathbb{1}) - |p)\right]. \tag{56}$$

Indeed, using the expectation values (51) and the conditions (54) imply

$$0 \leq \tfrac{1}{2}\left[1 - \langle p\rangle_z\right] \leq 1, \quad 0 \leq \tfrac{1}{2}\left[1 + \langle p\rangle_z\right] \leq 1, \quad |\langle A\rangle_z| \leq \tfrac{1}{2}\left[1 + \langle p\rangle_z\right]. \tag{57}$$

The first two conditions ensure that the probabilities $\tfrac{1}{2}[1 \pm \langle p\rangle_z]$ of being in the parity $\pm 1$ sector lie in the range $[0,1]$. The third condition ensures that conditional on being in the parity $+1$ sector, the difference of the individual occupations of its states $|\tau)$ cannot become too large: For a valid physical state in the parity $+1$ sector, the length of the (1-dimensional) Bloch vector $\langle A\rangle_z$ should not exceed the radius of the (1-dimensional) Bloch sphere set by the total probability $\tfrac{1}{2}[1 + \langle p\rangle_z]$ of being in that sector. Applying the duality mapping we find that the dual quantities also obey the corresponding constraints:

$$0 \leq \tfrac{1}{2}\left[1 - \langle p\rangle_{\bar{z}}\right] \leq 1, \quad 0 \leq \tfrac{1}{2}\left[1 + \langle p\rangle_{\bar{z}}\right] \leq 1, \quad |\langle A\rangle_{\bar{z}}| \leq \tfrac{1}{2}\left[1 + \langle p\rangle_{\bar{z}}\right]. \tag{58}$$

Thus, the dual and actual stationary vector are simultaneously legitimate states.

## 5.6 "Universal" stationary duality relation between expectation values

The result (51) expresses the stationary observables for the system and the dual system in the same set of (shifted-)duality invariants. This already implied the useful relation (44) but one can go one step further and, remarkably, express the *dual* polarization and parity in terms of the *actual* polarization and parity:

$$\langle \bar{A}\rangle_{\bar{z}} = -\langle A\rangle_{\bar{z}} = F\left(\langle A\rangle_z, \langle p\rangle_z\right) \cdot \langle A\rangle_z, \qquad \tfrac{1}{2}\left[1 + \langle p\rangle_{\bar{z}}\right] = F\left(\langle A\rangle_z, \langle p\rangle_z\right) \cdot \tfrac{1}{2}\left[1 + \langle p\rangle_z\right]. \tag{59a}$$

This *stationary duality relation* thereby explicitly expresses $|\bar{z})$ in terms of $|z)$ [Eq. 41a] by a rational function $F$ which is "universal" in the sense that it is *independent* of the physical parameters:

$$F\left(\langle A\rangle_z, \langle p\rangle_z\right) \equiv \frac{\tfrac{1}{2}\left[1 - \langle p\rangle_z\right]}{\tfrac{1}{2}\left[1 + \langle p\rangle_z\right] - \langle A\rangle_z^2}. \tag{59b}$$

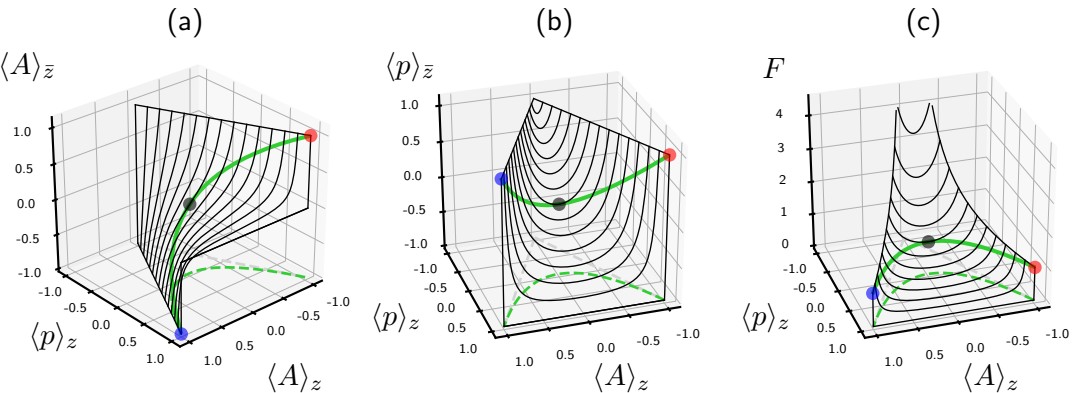

Figure 2: (a) Stationary polarization $\langle A \rangle_{\bar{z}}$ and (b) parity $\langle p \rangle_{\bar{z}}$ of the dual system as function of the stationary parity $\langle p \rangle_z$ for various polarizations $\langle A \rangle_z$ of the actual system [Eq. (59a)]. (c) Non-linear rescaling factor $F$ as function of $\langle A \rangle_z$ and $\langle p \rangle_z$ [Eq. (59b)]. The self-duality condition $F(\langle A \rangle_z, \langle p \rangle_z) = 1$ (green 3D curve) holds along the curve $\langle p \rangle_z = \langle A \rangle_z^2$ (dashed green 2D curve) in base plane which includes pure states $|\pm)$ (blue/red points) and the maximally mixed state (black point) which is the only state for which $|\bar{z}) = |z)$. Corresponding values are shown in (a) and (b).

The relation (59) is remarkable: given only polarization and parity *expectation values* of the actual system with physical parameters $(\epsilon, U, \alpha, \mu, T, \Gamma)$ it allows to compute these values for the system with dual physical parameters $(-\epsilon, -U, -\alpha, -\mu, T, \Gamma)$ without requiring any reference to these parameters: We plot in Fig. 2(a-b) the (components of the) dual stationary state as a *non-linear* function of the (components of the) stationary state of the actual system. In Fig. 2(c) we plot the scaling factor (59b) which takes non-negative values on the domain (57) of physically allowed values since there $\frac{1}{2}[1 + \langle p \rangle_z] \geq |\langle A \rangle_z| \geq \langle A \rangle_z^2$. It can be arbitrarily large (even diverging at $\langle p \rangle_z = -1$ and $\langle A \rangle_z = 0$) while always producing legitimate values $\langle p \rangle_{\bar{z}}, \langle A \rangle_{\bar{z}} \in [-1, 1]$. This is possible due to the non-linearity of the duality relation (59), i.e., the dependence of $F$ on $\langle A \rangle_z$ and $\langle p \rangle_z$.

The stationary duality (59) has been derived previously in Ref. [16] for master equations obeying both fermionic duality and detailed balance (which is the case here). This is highlighted in one of the two derivations of Eq. (59) given in App. A via the energy eigenbasis. However, here, in our duality-adapted basis this relation acquires a simpler form of a rescaling by a single factor $F(\langle A \rangle_z, \langle p \rangle_z)$ because we have written relation (59a) in terms of the probability $\frac{1}{2}[1 + \langle p \rangle_z]$ for the parity $+1$ sector and the polarization $\langle A \rangle_z$ in that sector. This nicely connects to our earlier result that the stationary state for the actual system $|z)$ and the dual system $|\bar{z})$ are simultaneously legitimate states: conditions (57) and (58) are equivalent since the factor $F$ drops out in $|\langle A \rangle_z|/[\frac{1}{2}[1 + \langle p \rangle_z]] = |\langle A \rangle_{\bar{z}}|/[\frac{1}{2}[1 + \langle p \rangle_{\bar{z}}]] \leq 1$: The duality mapping between stationary states preserves the magnitude of the (1-dimensional) Bloch vector relative to the (1-dimensional) Bloch sphere in the parity $+1$ sector.

Relation (59) also highlights the special situation of *self-duality* of stationary observables drawn in green in Fig. 2. This is defined as:

$$\langle \bar{A} \rangle_{\bar{z}} = -\langle A_{\bar{z}} \rangle = \langle A \rangle_z, \qquad\qquad \langle p \rangle_{\bar{z}} = \langle p \rangle_z. \qquad (60)$$

In this case the dual stationary state $|\bar{z})$ [Eq. (41a)] and actual one, $|z)$ [Eq. (42)], *differ*, but *only* by inverting the polarization. This is expected since the duality mapping inverts the energy spectrum, swapping the Andreev states $\overline{|\tau)} = |-\tau)$ and thus inverting the polarization

of their occupations. In this case $F(\langle A\rangle_z, \langle p\rangle_z) = 1$ which by Eq. (59b) is equivalent to having parity uniquely fixed by polarization in the stationary actual system and likewise in the dual system:

$$\langle p\rangle_z = \langle A\rangle_z^2, \qquad\qquad \langle p\rangle_{\bar{z}} = \langle A\rangle_{\bar{z}}^2. \qquad (61)$$

One verifies[7] that this is equivalent to self-duality of the charge decay rate, $\bar{\gamma}_c \equiv \gamma_p - \gamma_c = \gamma_c$, which is satisfied if and only if

$$\gamma_c = \tfrac{1}{2}\gamma_p, \qquad (62)$$

or $\gamma_C \equiv \gamma_c - \gamma_p/2 = 0$. This condition is satisfied at many parameter points, see Ref. [19], and for $T \to \infty$ it is even satisfied for all values of the remaining parameters, including $U$. More interestingly, at any finite $T$ for $U = 0$ self-duality always holds as we verify in App. B.1: By inspection of $\gamma_c = \frac{1}{2}\sum_{\eta\tau} W_{1,\tau}^{\eta}$ [Eq. (11),(49a)] considering all values of $\delta, \mu, \alpha, \Gamma$,

$$\gamma_c > \tfrac{1}{2}\gamma_p \Leftrightarrow U > 0, \qquad \gamma_c = \tfrac{1}{2}\gamma_p \Leftrightarrow U = 0, \qquad \gamma_c < \tfrac{1}{2}\gamma_p \Leftrightarrow U < 0. \qquad (63)$$

Thus, *zero interaction* is equivalent to requiring *exact* self-duality (60) for all values of these remaining parameters. One obtains approximate self-duality $\gamma_c \approx \frac{1}{2}\gamma_p$ asymptotically by rendering the interaction ineffective by making one energy scale dominate, e.g., large bias voltage $|\mu|$, strong pairing $|\alpha|$, large detuning $|\delta|$, or high temperature $T$, see App. B.2. In Secs. 6.4 and 7 we discuss the impact of self-duality on the transport in such cases. A system that is self-dual in the above sense ($U = 0$) necessarily has non-negative stationary parity $\langle p\rangle_z \geq 0$ [Eq. (61)] for any value of the remaining parameters as seen in Fig. 2. Strictly negative parity for some parameters thus requires nonzero magnitude of the interaction, $U \neq 0$. Such a violation of self-duality corresponds to a nonzero value of (+) invariant $\gamma_C = \gamma_c - \gamma_p/2$ [Eq. (63), cf. (55)] with the same sign as the interaction $U$.

As mentioned, self-duality (60) should not be confused with equality of the actual and dual stationary state, $|\bar{z}) = |z)$. The latter is a stronger condition and occurs when $\langle A\rangle_{\bar{z}} = \langle A\rangle_z$ and $\langle p\rangle_{\bar{z}} = \langle p\rangle_z$. This is satisfied only[8] for $\langle A\rangle_z = \langle p\rangle_z = 0$ implying $F = 1$. It is thus a special case of self-duality indicated by a black dot in Fig. 2 for which the stationary state is maximally mixed, $|z) = \frac{1}{4}|\mathbb{1})$, i.e., with uniform occupation of Andreev states ($z_\tau = 1/4$) and spin states ($z_1 = 1/2$, including spin-degeneracy). In addition to $\gamma_c = \gamma_p/2$ required by self-duality, this requires vanishing transition-rate asymmetry, $\gamma_s = 0$ [Eq. (51a)]. This again can be satisfied at many points[9], and holds trivially $T \to \infty$ for all parameters. Unlike self-duality, at $U = 0$ it does not hold for all parameters.

## 5.7 Transport current supercovectors

We can now complete the solution of the transport problem by deriving duality-adapted expressions for the currents. First, we express the left vector for the charge current (20a) in terms of the left eigenvectors of $W$. Since we again from the beginning choose a formulation

---

[7] Inserting Eq. (51) into Eq. (61) gives two solutions: $\gamma_c = \gamma_p/2$ and $|\gamma_s| = \gamma_c$. For the latter case the physical bound $|\gamma_s| \leq \gamma_p - \gamma_c$ [Eq. (54)] implies $\gamma_p/2 \geq \gamma_c$. Noting the bounds (63) we see that for $U \geq 0$ the lower-bound $\gamma_p/2 \leq \gamma_c$ implies $\gamma_p/2 = \gamma_c$, i.e., the second solution is a special case of the first one, whereas for $U < 0$ the strict upper-bound $\gamma_p/2 > \gamma_c$ rules it out.

[8] $\langle A\rangle_{\bar{z}} = \langle A\rangle_z$ in Eq. (59) implies $\langle A\rangle_{\bar{z}} = \langle A\rangle_z = 0$ since $F \geq 0$ and by Eq. (59b) $F = [1 - \langle p\rangle_z])/[1 + \langle p\rangle_z]$. Requiring $\langle p\rangle_{\bar{z}} = \langle p\rangle_z$ by Eq. (59) gives $[1 + \langle p\rangle_{\bar{z}}]/2 = [1 + \langle p\rangle_z]/2 = [1 - \langle p\rangle_z]/2$ so $\langle p\rangle_z = 0$.

[9] In particular, note that $\gamma_s = 0$ *can* hold at the often discussed "symmetry point" $\delta = 0$ ($\epsilon = -\frac{1}{2}U$) even when $U \neq 0$ *but it need not* hold, see our numerical analysis in Ref. [19].

in terms of scalar invariants and operators (anti)symmetric under the duality mapping, we find the very compact form

$$(I_N| = \sum_{\eta\tau} \eta\Big[ W_{1,\tau}^\eta(\tau| + 2W_{\tau,1}^\eta(1|\Big] = \gamma_c'(1| + \gamma_s'(A| \tag{64a}$$

$$= \big[\gamma_c' + \gamma_s'\langle A\rangle_z\big](z'| + \gamma_s'(c'|. \tag{64b}$$

Here we eliminated $W_{\tau,1}^\eta$ in favour of $W_{1,\tau}^\eta$ [Eq. (27)], inserted the decomposition (29) of the transport rates, changed basis using (35) and then rewrote in terms of the left eigenbasis (41). As anticipated, the primed ($=\eta$-antisymmetric) invariants $\gamma_c'$ and $\gamma_s'$ occur only in the transport quantities sensitive to electron transfer direction $\eta$.

For the energy and heat current one can proceed analogously[10] by rewriting the left supervector for the energy current (20b) into the metal. However, it is more convenient to start from the energy loss of the proximized dot [9],

$$I_E(t) = -\partial_t\langle H_D\rangle(t) = -(H_D|W|\rho(t)) = (I_E|\rho(t)), \tag{65}$$

as the Cooper pair condensate does not contribute to the energy of the proximized dot, whereas it does contribute to its charge. Inserting the energies (3) and (5) and performing the same steps, reveals the key feature of the energy, namely, that it couples to interaction $U$ via the parity:

$$(H_D| = 2E_1(1| + \sum_\tau E_\tau(\tau| = \tfrac{1}{2}\delta_A(A| + \tfrac{1}{4}U(p| + \big[\dots\big](1| \tag{66a}$$

$$= \tfrac{1}{2}\big(\delta_A - U\langle A\rangle_{\bar z}\big)(c'| + U(p'| + \big[\dots\big](z'|. \tag{66b}$$

The terms with coefficients $[\dots]$ are diagonal in the energy basis but don't contribute to the energy current since $(z'|W = 0$. Thus, $(I_E| = -(H_D|W$ is given by

$$(I_E| = \tfrac{1}{2}\big(\delta_A - U\langle A\rangle_{\bar z}\big)\gamma_c(c'| + U\gamma_p(p'|. \tag{67}$$

The heat current $I_Q(t) = I_E(t) - \mu I_N(t) = (I_Q|\rho(t))$ is obtained from

$$(I_Q| = -\mu\big[\gamma_c' + \gamma_s'\langle A\rangle_z\big](z'| + \Big[\tfrac{1}{2}\big(\delta_A - U\langle A\rangle_{\bar z}\big)\gamma_c - \mu\gamma_s'\Big](c'| + U\gamma_p(p'|. \tag{68}$$

## 6 General analytical result

The main results of the paper are the final analytical formula for the state evolution

$$|\rho(t)) = \Big\{\tfrac{1}{4}|1) + \langle A\rangle_z\tfrac{1}{2}|A) + \langle p\rangle_z\tfrac{1}{4}|p)\Big\} + \tfrac{1}{2}\Big[|A) - \langle A\rangle_{\bar z}|p)\Big]e^{-\gamma_c t}\big[\langle A\rangle_{\rho_0} - \langle A\rangle_z\big]$$
$$+ |p)e^{-\gamma_p t}\Big\{\tfrac{1}{4}\big[\langle p\rangle_{\rho_0} - \langle p\rangle_z\big] + \tfrac{1}{2}\langle A\rangle_{\bar z}\big[\langle A\rangle_{\rho_0} - \langle A\rangle_z\big]\Big\}, \tag{69}$$

and the formulas for the charge and heat current,

$$I_N(t) = \big(\gamma_c' + \gamma_s'\langle A\rangle_z\big) + \gamma_s'e^{-\gamma_c t}\big[\langle A\rangle_{\rho_0} - \langle A\rangle_z\big], \tag{70}$$

$$I_Q(t) = -\mu\big\{\gamma_c' + \gamma_s'\langle A\rangle_z\big\} + \Big\{\tfrac{1}{2}\big(\delta_A - U\langle A\rangle_{\bar z}\big)\gamma_c - \mu\gamma_s'\Big\}e^{-\gamma_c t}\big[\langle A\rangle_{\rho_0} - \langle A\rangle_z\big]$$
$$+ U\gamma_p e^{-\gamma_p t}\Big\{\tfrac{1}{4}\big[\langle p\rangle_{\rho_0} - \langle p\rangle_z\big] + \tfrac{1}{2}\langle A\rangle_{\bar z}\big[\langle A\rangle_{\rho_0} - \langle A\rangle_z\big]\Big\}, \tag{71}$$

---

[10]Eq. (9) can be written $I_E(t) = -\sum_{\eta\tau} E_{+,\tau}\big[-W_{1,\tau}^\eta\rho_\tau(t) + W_{\tau,1}^\eta\rho_1(t)\big] = (I_E|\rho(t))$ with left vector $(I_E| = -\sum_{\eta\tau} E_{+,\tau}\big[-W_{1,\tau}^\eta(\tau| + 2W_{\tau,1}^\eta(1|\big]$ and one then proceeds from there.

where the energy current $I_E(t)$ is given by setting $\mu = 0$ in the expression for $I_Q(t)$. This solution applies to any initial state $|\rho_0)$ that is diagonal in the energy basis, specified by the initial expectation values of the polarization $\langle A_0 \rangle_{\rho_0}$ and parity $\langle p \rangle_{\rho_0}$ subject to the constraint (54), $|\langle p \rangle_{\rho_0}| \leq 1$ and $|\langle A_0 \rangle_{\rho_0}| \leq \frac{1}{2}[1 + \langle p \rangle_{\rho_0}]$, guaranteeing that the initial state is physical. where by $A_0$ we indicate the dependence of polarization on initial parameters. This is the solution to the time-evolution and transport problem of Ref. [36]. In the following we mention its salient features, discussing some specific limiting cases analytically in Sec. 7, thereby preparing for the numerical analysis of the general case in Ref. [19].

## 6.1 State evolution

The state supervector (69) is obtained by inserting the amplitude functions (53) into Eq. (52). The key advantage of this duality-based solution is that we obtain the state-evolution *entirely* in terms of *stationary* expectation values of the appropriate observables $p$ and $A$, in particular the all-important amplitudes of the transient decay with rates $\gamma_c$ and $\gamma_p$. Roughly speaking, if we know the initial and stationary values of these observables we understand the entire intermediate state evolution. Here, the amplitudes are expressed using result (53d) as the difference between initial and stationary value of the observables $A$ and $p$, their "initial excess". This makes explicit that, as expected, $|\rho(t)) = |\rho_0)$ for all $t \geq 0$ if $|\rho_0) = |z)$ unlike the form (53c) used in Ref. [9], where the "excess" of observables with respect to their stationary value does not explicitly enter see Sec. 7.2.

As in prior work we find that the relevant stationary values are *not* only those of the actual system but also those of the *dual* system. This remarkable insight has significantly advanced the analysis of the parameter dependence of the transient state evolution [9,16,19,54]. Our result extends this to interacting systems proximized by a superconductor but additionally points out two important refinements.[11] It is well known that a time-evolving quantum state can be expanded in a set of fixed observables (generalized Bloch expansion) with coefficients which are their *time-dependent* averages in that state. Here we instead express the time-dependent state in averages of the *stationary* state ($t \rightarrow \infty$), the initial state, and decay exponentials.

First, the relation $(p'|\rho_0) = (p\bar{z}|\rho_0)$ [Eq. (40)] indicates that the amplitude of the $\gamma_p$-decay depends on how well the initial state $|\rho_0)$ "compares" with the *dual* stationary state $|\bar{z})$ as discussed in Ref. [9, 15]. The above mentioned form (53c) of this amplitude suggests that one needs to analyse the dual stationary parity $\langle p \rangle_{\bar{z}}$ for this. However, our result (53d) explicitly shows that the dual stationary state enters *only* through its polarization $\langle A \rangle_{\bar{z}}$ and *only* provided that there is an initial polarization excess, $\langle A \rangle_{\rho_0} \neq \langle A \rangle_z$. This simplifies the analysis and these results carry over to the case $\alpha = 0$, see Sec. 7.1.

Second, we can go one step further and substitute the stationary duality relation (59a) into Eq. (69): we see that the amplitudes of the transient decay with rates $\gamma_c$ and $\gamma_p$ ultimately depend only on the *stationary* expectation values of $A$ and $p$ of the *system alone*. Thus, apart from the decay rates the *evolution towards* the stationary state can be understood in detail by the stationary state *itself*. If anything, this *non-linear* dependence on its stationary state (59b) highlights the non-trivial nature of the transient evolution. It does not invalidate the usefulness of above mentioned analysis via dual stationary values which enter linearly: One either analyses the actual stationary system and the dual parity or one analyses only the actual stationary system but uses the non-linear –but parameter independent– relation (59).

---

[11]Equation (69) is thus more than a rewriting of the expression for $|\rho(t))$ given in Ref. [38], see Eq. (20-25) of that work for the special case of zero transverse Bloch vector ($I_x = I_y = 0$).

## 6.2 Transport evolution

The transport currents (70)-(71) follow by taking the scalar product of the state (69) with the corresponding left current supervectors [Eqs. (64b) and (68)] and using bi-orthonormality of the eigen-supervectors. This leads to the physically appealing structure

$$I_x(t) = (I_x|\rho(t)) = (I_x|z) + (I_x|c)e^{-\gamma_c t}(c'|\rho_0) + (I_x|p)e^{-\gamma_p t}(p'|\rho_0), \tag{72}$$

for $x = N, E, Q$. Denoting $\gamma_z \equiv 0$, each term $y = z, c, p$ contributing to a transport quantity $I_x(t)$ on time-scale $\gamma_y^{-1}$ depends on two factors $(I_x|y)$ and $(y'|\rho_0)$. The first factor quantifies the ability of quantity $x$ to "probe" the $y$-part of the state evolution, the second one captures the initial-state dependence of the latter, see also Ref. [54]. For example, the charge current (70) does not probe the parity decay, $(I_N|p) = 0$, and thus –by duality– depends on the physical parameters only through the stationary state of the actual system, in particular, through the value of the polarization observable $\langle A \rangle_z$ in $(c'|\rho_0)$ [Eq. (53b)]. By contrast, the energy and heat current do probe the parity decay, $(I_E|p), (I_Q|p) \neq 0$, and thus both *linearly* depend on the dual stationary state through $\langle A \rangle_{\bar{z}}$. (As discussed at the end of the previous section, this dependence can ultimately be brought back to $\langle A \rangle_z$ and $\langle p \rangle_z$ via relation (59) but only at the price of *non-linear* dependence.)

As for the state evolution, *non-trivial* time-dependence of the transport –which is supposed to probe the state of the system– is expressed explicitly in terms of the stationary state, i.e., the values of observables $p$ and $A$ that characterize it. In the case of energy transport (71) the additional pre-factors $\delta$, $U$ and $\mu$ that enter do not complicate the analysis. For the charge transport (70) the charge-sensitive coefficients $\gamma'_s$ and $\gamma'_c$ do depend non-trivially on the parameters, but their dependence is again restricted by duality: As we show below in Sec. 7, their parameter dependence is closely related to that of $\gamma_c$ and $\gamma_s$, respectively.

Finally, we note that the explicit dependence of the heat current (71) on the excess polarization and parity can be used to separately measure its different contributions provided one has full control over the initial state. For example, for an initial state with no excess polarization, $\langle A \rangle_{\rho_0} = \langle A \rangle_z$, only the second term in Eq. (71) contributes with single-exponential decay with rate $\gamma_p$. By contrast, for an initial state with no excess parity, $\langle p \rangle_{\rho_0} = \langle p \rangle_z$, the decay is still double exponential with rates $\gamma_c$ and $\gamma_p$.

## 6.3 Components of duality invariants

We now illustrate how the fermionic duality can be used to investigate the solutions (69)-(71) in more detail by decomposing the (shifted-) duality invariants into "components"

$$\gamma_C = \kappa_C + \frac{\delta}{\delta_A}\kappa'_s, \qquad \gamma_c = \kappa_c + \frac{\delta}{\delta_A}\kappa'_s, \qquad \gamma'_c = \kappa'_c + \frac{\delta}{\delta_A}\kappa_s, \tag{73a}$$

$$\gamma_s = \kappa_s + \frac{\delta}{\delta_A}\kappa'_c, \qquad \gamma'_s = \kappa'_s + \frac{\delta}{\delta_A}\kappa_c, \qquad \gamma'_S = \kappa'_s + \frac{\delta}{\delta_A}\kappa_C, \tag{73b}$$

while still avoiding the more cumbersome full expressions of the rates. The decomposition (73) separates the contributions from the two summands in $\Gamma_\pm = \gamma_p(1 \pm \delta/\delta_A)/2$ [Eq. (13)] which enter the invariants via Eq. (11): Formally each component is obtained by replacing $\Gamma_{\eta\tau} \to \frac{1}{2}\gamma_p$ in the rate-expression of the corresponding invariant (49) after inserting Eq. (11). Explicitly,

$$\kappa_c = \frac{1}{4}\gamma_p \sum_{\eta\tau} f^{-\eta}(E_{\eta,\tau} - \mu), \qquad \kappa'_c = \frac{1}{4}\gamma_p \sum_{\eta\tau} \eta f^{-\eta}(E_{\eta,\tau} - \mu), \tag{74a}$$

$$\kappa_s = \frac{1}{4}\gamma_p \sum_{\eta\tau} \tau f^{-\eta}(E_{\eta,\tau} - \mu), \qquad \kappa'_s = \frac{1}{4}\gamma_p \sum_{\eta\tau} \tau \eta f^{-\eta}(E_{\eta,\tau} - \mu). \tag{74b}$$

That these are appropriate variables is established by the fact that they transform in the same simple way as the corresponding invariants [Eqs. (30), (46)]:

$$\kappa_C = -\bar{\kappa}_C \,, \qquad\qquad \kappa_c = \gamma_p - \bar{\kappa}_c \,, \qquad\qquad \kappa_s = \bar{\kappa}_s \,, \qquad (75a)$$

$$\kappa'_c = \quad -\bar{\kappa}'_c \,, \qquad\qquad \kappa'_s = \bar{\kappa}'_s \,. \qquad (75b)$$

Like the four (shifted-)invariants that they decompose, the behaviour of these four components as function of the physical parameters provides a complete understanding of the time-dependence of the solution of the problem. The prime notation for the components makes the same physical distinction as for the invariants [a component has (no) prime if it determines charge transfer (state evolution)] which is, however, reversed when occurring with prefactor $\delta/\delta_A$. For example, primed invariants $\gamma'_c, \gamma'_s$ [Eq. (64a)] associated with transport count charge transfer $\eta$ with effective rates $\Gamma_{\eta\tau} = \gamma_p(1 + \eta\tau\delta/\delta_A)/2$: a transport component is thus either $\eta$-antisymmetric or $\eta$-symmetric with factor $\delta/\delta_A$.

The components thus have the advantage that they cleanly separate all resonant behaviour of the superconductor (explicit $\delta$ dependence relative to $\mu_S = 0$ through $\Gamma_{\eta\tau}$) from the superconductor's indirect effect via the Andreev levels (implicit $\delta$ dependence relative to $\mu$ through $f^{-\eta}(E_{\eta,\tau})$ in Eq. (74)). By doing so, the decomposition (73) shows that the pairs of duality invariants $\gamma_c$, $\gamma'_s$ and $\gamma'_c$, $\gamma_s$, respectively, playing distinct physical roles, actually have similar behaviour since they depend on the same components which can mix up and cancel out when simplifying final expressions at the price of losing the physical distinction. This is nicely illustrated for the expression for stationary charge current $I_N(\infty) = \gamma'_c + \gamma'_s \langle A \rangle_z$ given by the first term in Eq. (70):

$$I_N(\infty) = \frac{\gamma'_c \gamma_c - \gamma'_s \gamma_s}{\gamma_c} = \frac{\kappa_c \kappa'_c - \kappa_s \kappa'_s}{\kappa_c + \frac{\delta}{\delta_A}\kappa'_s}\left(1 - \frac{\delta^2}{\delta_A^2}\right). \qquad (76)$$

We automatically extract a resonant Lorentzian $\delta$-dependence reflecting the non-equilibrium Cooper pair transport to the superconductor while its non-trivial modification by the Andreev states is contained in the pre-factor governed by the components of the invariants. The latter is a joint effect of both transport and state-evolution invariants. This stationary current was studied in Refs. [23, 34–36] in the weak-coupling limit as well as in Refs. [55–58] in the limit of strong coupling where analytical results were limited to specific regimes. Note that the stationary energy current equals zero, $I_E(\infty) = 0$.

## 6.4 Self-duality and transient transport

In the limiting cases discussed in Sec. 7 the state and transport evolutions simplify substantially when the system obeys self-duality (60), i.e., $\gamma_c = \frac{1}{2}\gamma_p$ [Eq. (62)]. As discussed in Sec. 5.6, this can either hold exactly for all applied voltages ($U = 0$) or asymptotically (e.g., $\mu \to \infty$). We can then generically simplify the stationary state (41a) by eliminating parity in favour of polarization [Eq. (61)]:

$$|z) = \frac{1}{4}|\mathbb{1}) + \langle A \rangle_z \frac{1}{2}|A) + \langle A \rangle_z^2 \frac{1}{4}|p) = \sum_\tau |\tau)\frac{1}{4}\left[1 + \tau\langle A \rangle_z\right]^2 + |1)\frac{1}{2}\left[1 - \langle A \rangle_z^2\right]. \qquad (77)$$

In the evolving state (69) this is not possible without further consideration since self-duality need not hold initially, $\langle p \rangle_{\rho_0} \neq \langle A_0 \rangle_{\rho_0}^2$, because the energy mixture $\rho_0$ is arbitrary. However, as discussed in Ref. [19], a relevant class of initial states are those prepared from a stationary state of the system at some different gate voltage $\delta_0$ –which by the above assumptions is still self-dual– and thus $\langle p \rangle_{z_0} = \langle A_0 \rangle_{z_0}^2$. This way one obtains an initial state with $\langle p \rangle_{\rho_0} = \langle p \rangle_{z_0}$ and

$\langle A \rangle_{\rho_0} = \theta \langle A_0 \rangle_{z_0}$ where $\theta \in [-1,1]$ captures details of the initialization [19]. In this case all decay amplitudes in the evolving state are fixed by the stationary and initial polarization:

$$|\rho(t)) = \tfrac{1}{4}|\mathbb{1}) + \langle A \rangle_z \tfrac{1}{2}|A) + \langle A \rangle_z^2 \tfrac{1}{4}|p) + \tfrac{1}{2}\big[|A) + \langle A \rangle_z |p)\big] e^{-\frac{1}{2}\gamma_p t}\big[\theta \langle A_0 \rangle_{z_0} - \langle A \rangle_z \big]$$
$$+ |p) e^{-\gamma_p t}\big\{ \tfrac{1}{4}\langle A_0 \rangle_{z_0}^2 + \tfrac{1}{4}\langle A \rangle_z^2 - \tfrac{1}{2}\theta \langle A \rangle_z \langle A_0 \rangle_{z_0} \big\}. \tag{78}$$

Moreover, the coefficient of the parity contribution can be verified to be non-negative which in general [Eq. (53c)-53d] need not be true even when accounting for constraints (57)-(58). The currents likewise simplify,

$$I_N(t) = \big(\gamma_c' + \gamma_s'\langle A \rangle_z\big) + \gamma_s' e^{-\frac{1}{2}\gamma_p t}\big[\theta \langle A_0 \rangle_{z_0} - \langle A \rangle_z \big], \tag{79}$$

$$I_Q(t) = -\mu\big\{\gamma_c' + \gamma_s'\langle A \rangle_z\big\} + \big\{\tfrac{1}{2}\delta_A \gamma_c - \mu \gamma_s'\big\} e^{-\frac{1}{2}\gamma_p t}\big[\theta \langle A_0 \rangle_{z_0} - \langle A \rangle_z \big]. \tag{80}$$

In particular $I_Q(t)$ has no parity contribution, either exactly (for $U = 0$) or asymptotically (for large $\mu$ where it is dominated by the terms $\propto \mu$, in which case $\tfrac{1}{2}\delta_A \gamma_c$ must also be dropped). Finally, whenever we have stationary transport, $\alpha \neq 0$, self-duality $\gamma_c = \gamma_p/2$, is equivalent to having a constant component, $\kappa_c = \gamma_p/2$, and a vanishing one, $\kappa_s' = 0$ [Eq. (49), (73)]. The stationary current (76) is then modulated by just one component:

$$I_N(\infty) = \gamma_c' + \gamma_s'\langle A \rangle_z = \kappa_c' \times \left(1 - \frac{\delta^2}{\delta_A^2}\right). \tag{81}$$

# 7 Limiting cases

Finally, to highlight various physical effects captured by the duality invariants we present explicit formulas for three different limiting cases, taking either $\alpha \to 0$, $U \to 0$ or $\mu \to \infty$.

## 7.1 Interacting dot without pairing ($\alpha = 0$)

For a quantum dot coupled only to a normal metal ($\alpha = 0$) the fermionic duality expresses a symmetry of the problem which is not at all obvious in the presence of strong interaction ($U \neq 0$) [9, 14]. It is instructive to see how this case is recovered noting that by the assumption (6) we *cannot* simply send $\alpha \to 0$ within the range of applicability of the master equation (7) derived in Ref. [36]. However, for $\Gamma \ll |\alpha| \ll T$ temperature fluctuations eradicate every effect of the pairing except for a gate voltage regime around the resonance with the superconductor, $|\delta| \lesssim |\alpha|$, which becomes vanishingly small as $\alpha \to 0$.

Thus, in this limit our general solution should coincide with that obtained for $\alpha = 0$ in Ref. [9] at every $\delta \neq 0$. Indeed, when closing the pairing-induced gap $\alpha \to 0$ the transport- and transition rates become discontinuous functions of the gate voltage $\delta$ through both the transition energies (12) and the effective rates (13). The origin of the discontinuity lies in Eq. (3) where for $\alpha \neq 0$ we label the eigenenergies $H_D$ such that they correspond to $\delta$-continuous branches of the square root. Correspondingly, the labelling of eigenstates (4) becomes a discontinuous choice of labelling the obvious eigenstates at $\alpha = 0$:

$$|\tau\rangle\langle\tau| \overset{\alpha \to 0}{=} |0\rangle\langle 0| \text{ for } \tau = -\frac{\delta}{|\delta|} \quad and \quad |\tau\rangle\langle\tau| \overset{\alpha \to 0}{=} |2\rangle\langle 2| \text{ for } \tau = +\frac{\delta}{|\delta|}. \tag{82}$$

This gate-voltage dependence of the states is essential to the problem with superconducting pairing $\alpha \neq 0$[12] and there is no reason to proceed otherwise. However, for $\alpha \to 0$ no physical

---

[12]This gate-voltage dependence warrants the careful discussion of the state initialization in Ref. [19].

result at $\delta \neq 0$ should depend on the choice of labelling made for $\alpha \neq 0$ but verifying this is messy since *all* transport rates (11) become discontinuous for $\alpha \to 0$. The duality invariants now have the nice property that for $\alpha \to 0$ they are either continuous or discontinuous only by the *sign factor* $\delta/|\delta|$. Duality thus automatically either cancels all discontinuities or collects them into a single factor. This is clearly useful for numerical calculations but also simplifies the analysis of the limiting behaviour of proximized quantum dots here.

Indeed, for $\alpha \to 0$ the invariants $\gamma_s$, $\gamma'_s$ associated with an asymmetry with respect to energy eigenstates [$\tau$ branches, see Eq. (49a)-(49b)] are discontinuous and are related by the $\delta$-sign to continuous invariants $\gamma_c$, $\gamma'_c$ which are ignorant of this asymmetry:

$$\gamma_s \overset{\alpha \to 0}{=} \frac{\delta}{|\delta|}\gamma'_c, \qquad \gamma'_s \overset{\alpha \to 0}{=} \frac{\delta}{|\delta|}\gamma_c. \tag{83}$$

Indeed, in Eqs. (49a)-(49b) only the term $\tau = \eta \delta/|\delta|$ contributes to invariants $\gamma_c$, $\gamma'_c$, giving continuous functions of the transition energies $E_{\eta,\eta} = \frac{1}{2}(\delta + \eta U) = \epsilon, \epsilon + U$ for $\eta = \mp$: [Eq. (12)]:

$$\gamma_c = \frac{1}{2}\gamma_p \sum_\eta f^{-\eta}(E_{\eta,\eta} - \mu), \qquad \gamma'_c = \frac{1}{2}\gamma_p \sum_\eta \eta f^{-\eta}(E_{\eta,\eta} - \mu). \tag{84}$$

Similarly, the parameter-dependent polarization observable [cf. Eq. (34)] is discontinuous in $\delta$ but in a simple way, see Eq. (82): with the charge polarization $N - \mathbb{1} = -|0\rangle\langle 0| + |2\rangle\langle 2|$

$$A = \sum_\tau \tau |\tau\rangle\langle\tau| \overset{\alpha \to 0}{=} \frac{\delta}{|\delta|}(N - \mathbb{1}). \tag{85}$$

Inserting Eq. (83) and Eq. (85) into the observables (43) we recover the continuous expressions of Ref. [9] but in the more compact form of duality invariants,

$$\langle N - \mathbb{1}\rangle_z \overset{\alpha \to 0}{=} -\frac{\gamma'_c}{\gamma_c}, \qquad \langle p\rangle_z \overset{\alpha \to 0}{=} 1 - 2\frac{\gamma_c^2 - \gamma_c'^2}{\gamma_p \gamma_c}, \tag{86a}$$

$$\langle N - \mathbb{1}\rangle_{\bar{z}} \overset{\alpha \to 0}{=} \frac{\gamma'_c}{\gamma_p - \gamma_c}, \qquad \langle p\rangle_{\bar{z}} \overset{\alpha \to 0}{=} 1 - 2\frac{(\gamma_p - \gamma_c)^2 - \gamma_c'^2}{\gamma_p(\gamma_p - \gamma_c)}. \tag{86b}$$

Thus, for "$\alpha \to 0$" our appropriate duality-adapted observables essentially reduce to the parity $p$ and the charge polarization $N - \mathbb{1}$ and our four duality invariants simplify to just two invariants $\gamma_c$ (state) and $\gamma'_c$ (transport) both continuous in gate voltage $\delta$. This result shows that also in Ref. [9] it is possible to eliminate all dual stationary observables in favour of the actual ones. Indeed, the stationary duality relation (59) also holds with the substitution $A \to N - \mathbb{1}$ (the $\delta$-signs cancel) and shows the consistency of the actual and dual positivity constraints $|\langle N - \mathbb{1}\rangle_z| \leq \frac{1}{2}[1 + \langle p\rangle_z]$ and $|\langle N - \mathbb{1}\rangle_{\bar{z}}| \leq \frac{1}{2}[1 + \langle p\rangle_{\bar{z}}]$.

In the state evolution (69) the $\delta$-signs cancel between duality-adapted observables and their expectation values recovering the continuous Eqs. (S3-S5) of Ref. [9] for $\alpha \to 0$:

$$|\rho(t)) \overset{\alpha \to 0}{=} \left[\frac{1}{4}|\mathbb{1}) + \langle N - \mathbb{1}\rangle_z \frac{1}{2}|N - \mathbb{1}) + \langle p\rangle_z \frac{1}{4}|p)\right] + \frac{1}{2}\left[|N - \mathbb{1}) - \langle N - \mathbb{1}\rangle_{\bar{z}}|p)\right]e^{-\gamma_c t}\left[\langle N\rangle_{\rho_0} - \langle N\rangle_z\right]$$
$$+ |p)e^{-\gamma_p t}\left\{\frac{1}{4}\left[\langle p\rangle_{\rho_0} - \langle p\rangle_z\right] + \frac{1}{2}\langle N - \mathbb{1}\rangle_{\bar{z}}\left[\langle N\rangle_{\rho_0} - \langle N\rangle_z\right]\right\}. \tag{87}$$

Likewise, for the currents (70)-(71) we recover Eq. (S8) of Ref. [9] with continuous factors:

$$I_N(t) \overset{\alpha \to 0}{=} \gamma_c e^{-\gamma_c t}\left[\langle N\rangle_{\rho_0} - \langle N\rangle_z\right], \tag{88}$$

$$I_Q(t) \overset{\alpha \to 0}{=} \left[\frac{1}{2}\left(\delta - U\langle N - \mathbb{1}\rangle_{\bar{z}}\right) - \mu\right]\gamma_c e^{-\gamma_c t}\left[\langle N\rangle_{\rho_0} - \langle N\rangle_z\right]$$
$$+ U\gamma_p e^{-\gamma_p t}\left\{\frac{1}{4}\left[\langle p\rangle_{\rho_0} - \langle p\rangle_z\right] + \frac{1}{2}\langle N - \mathbb{1}\rangle_{\bar{z}}\left[\langle N\rangle_{\rho_0} - \langle N\rangle_z\right]\right\}. \tag{89}$$

These expressions have been successfully applied to the analysis of both repulsive ($U > 0$) [9] and attractive ($U < 0$) quantum dots [14] in prior works to which we refer for concrete worked-out examples and discussion of experimental protocols. Compared to the result of Ref. [9] the last term in Eqs. (87) and (89) has been simplified using Eq. (44) and (53d). This makes explicit that without initial excess charge ($\langle N \rangle_{\rho_0} = \langle N \rangle_z$) the heat current exhibits strictly single-exponential $\gamma_p$ decay, whereas without initial excess parity ($\langle p \rangle_{\rho_0} = \langle p \rangle_z$) one still has double exponential heat decay, cf. Sec. 6.2.

A final point to note is that the distinct behaviour $\langle N \rangle_z$ and $\langle p \rangle_z$ is due to the interaction: only when additionally sending $U \to 0$ do we have $\gamma_c \to \frac{1}{2}\gamma_p$ since $E_{\eta,\eta} = \epsilon$ independent of $\eta$. By Eq. (62) the system is then self-dual such that the considerations of Sec. 6.4 apply: the parity is completely fixed by the charge-polarization

$$-\langle N - \mathbb{1} \rangle_{\bar{z}} \stackrel{\alpha, U \to 0}{=} \langle N - \mathbb{1} \rangle_z, \qquad \langle p \rangle_z \stackrel{U, \alpha \to 0}{=} \langle N - \mathbb{1} \rangle_z^2, \qquad \langle p \rangle_{\bar{z}} \stackrel{U, \alpha \to 0}{=} \langle N - \mathbb{1} \rangle_{\bar{z}}^2. \tag{90}$$

The other nonzero invariant simplifies to an antisymmetric step function, $\gamma_c' \to \frac{1}{2}\gamma_p[1 - 2f(\epsilon - \mu)]$ which determines the charge-polarization [Eq. (86a)]:

$$\langle N - \mathbb{1} \rangle_z \stackrel{\alpha, U \to 0}{=} 2f(\epsilon - \mu) - 1. \tag{91}$$

## 7.2 Proximized dot without interaction ($U = 0$)

For a non-interacting dot ($U = 0$) proximized by a superconductor ($\alpha \neq 0$), see Ref. [59], fermionic duality also remains a dissipative symmetry but its implications in this case have not been considered. Since in this case $\gamma_c$ has one vanishing component, $\kappa_s' \to 0$, and the other is constant, $\kappa_c \to \gamma_p/2$, the invariants again simplify:

$$\gamma_c \stackrel{U \to 0}{=} \frac{1}{2}\gamma_p, \qquad \gamma_s' \stackrel{U \to 0}{=} \frac{1}{2}\gamma_p \frac{\delta}{\delta_A}, \tag{92}$$

and we have *exact* self-duality by Eq. (62) and Sec. 6.4 again applies. Here this happens since $E_{\eta,\tau} \to \frac{1}{2}\eta\tau\delta_A$ depends only on the product $\eta\tau \equiv \lambda = \pm$ unlike the interacting case [Eq. (12)]. For the two remaining invariants $\gamma_c' = \kappa_c' + \frac{\delta}{\delta_A}\kappa_s$ and $\gamma_s = \kappa_s + \frac{\delta}{\delta_A}\kappa_c'$ the components read

$$\kappa_s \stackrel{U \to 0}{=} \frac{1}{2}\gamma_p \sum_{\lambda\eta} \frac{1}{2}\lambda\eta f^{-\eta}(\frac{1}{2}\lambda\delta_A - \mu), \qquad \kappa_c' \stackrel{U \to 0}{=} \frac{1}{2}\gamma_p \sum_{\lambda\eta} \frac{1}{2}\eta f^{-\eta}(\frac{1}{2}\lambda\delta_A - \mu). \tag{93}$$

Importantly, for $U = 0$ the self-duality (92) holds for all values of the remaining parameters $\delta, \mu, T, \Gamma$, and in particular, the pairing $\alpha$, such that we have according to Eq. (51)

$$\langle A \rangle_z \stackrel{U \to 0}{=} \langle \bar{A} \rangle_{\bar{z}}, \qquad \langle p \rangle_z \stackrel{U \to 0}{=} \langle p \rangle_{\bar{z}}, \qquad \langle p \rangle_z \stackrel{U \to 0}{=} \langle A \rangle_z^2, \qquad \langle p \rangle_{\bar{z}} \stackrel{U \to 0}{=} \langle A \rangle_{\bar{z}}^2. \tag{94}$$

Applying all simplifications (77)-(81) we obtain for the stationary polarization

$$\langle A \rangle_z \stackrel{U \to 0}{=} -2\frac{\gamma_s}{\gamma_p} = -\sum_\lambda \frac{1}{2}\left(\lambda + \frac{\delta}{\delta_A}\right) \sum_\eta \eta f^{-\eta}\left(\frac{1}{2}\lambda\delta_A - \mu\right), \tag{95}$$

and for the non-interacting stationary current (81):

$$I_N(\infty) = \gamma_c' + \gamma_s'\langle A \rangle_z = \frac{1}{2}\gamma_p \frac{1}{2}\sum_{\lambda\eta} \eta f^{-\eta}\left(\frac{1}{2}\lambda\delta_A - \mu\right) \times \left(1 - \frac{\delta^2}{\delta_A^2}\right). \tag{96}$$

### 7.3 Proximized dot at high bias ($\mu \gg |\alpha|, U, |\delta|, T, \Gamma$)

Finally, for an interacting and proximized dot at high bias, $\mu \gg U, |\alpha|, |\delta|, T, \Gamma$ (for simplicity denoted as "$|\mu| \to \infty$") we have *approximate* self-duality as mentioned in Sec. 6.4. In this case $|\mu| \gg |E_{\eta,\tau}|$ such that all dependence on the Andreev energies (and thus on $\eta$ and $\tau$) drops out in the components (74). Therefore the two components $\kappa_s, \kappa'_s \to 0$ vanish, one is constant $\kappa_c \to \gamma_p/2$ and one simplifies to $\kappa'_c \to \frac{1}{2}\gamma_p \sum_\eta \eta f^{-\eta}(-\mu) = \frac{1}{2}\gamma_p \tanh[\mu/(2T)]$. Once more the invariants have a simple property

$$\gamma_s \stackrel{|\mu|\to\infty}{=} \frac{\delta}{\delta_A}\gamma'_c, \qquad\qquad \gamma'_s \stackrel{|\mu|\to\infty}{=} \frac{\delta}{\delta_A}\gamma_c, \tag{97}$$

which is very similar to the $\alpha \to 0$ case (83) except for the fact that the "sign function" $\delta/|\delta|$ in Eq. (92) gets "broadened" by $\alpha$ in $\delta/\delta_A$. Note that this gate-voltage dependence of the pre-factor, originating from the state-dependence of the effective rate $\Gamma_{\eta\tau}$ [Eq. (28)], is not wiped out by the high bias which by our assumptions has to remain below the (infinite) gap of the superconductor, see Sec. 2. The remaining invariants read

$$\gamma_c \stackrel{|\mu|\to\infty}{=} \frac{1}{2}\gamma_p, \qquad\qquad \gamma'_c \stackrel{|\mu|\to\infty}{=} \frac{1}{2}\gamma_p \tanh\left(\frac{\mu}{2T}\right), \tag{98}$$

such that by Eq. (62) we again have self-duality (60):

$$\langle A \rangle_z \stackrel{|\mu|\to\infty}{=} \langle \bar{A} \rangle_{\bar{z}}, \qquad \langle p \rangle_z \stackrel{|\mu|\to\infty}{=} \langle p \rangle_{\bar{z}}, \qquad \langle p \rangle_z \stackrel{|\mu|\to\infty}{=} \langle A \rangle_z^2, \qquad \langle p \rangle_{\bar{z}} \stackrel{|\mu|\to\infty}{=} \langle A \rangle_{\bar{z}}^2. \tag{99}$$

and the simplifications (77)-(81) apply. Now the stationary polarization consists of factors independently controlled by the applied voltages:

$$\langle A \rangle_z \stackrel{|\mu|\to\infty}{=} -\frac{\delta}{\delta_A}\tanh\left(\frac{\mu}{2T}\right). \tag{100}$$

Likewise, in the stationary current (76) the pre-factor $\kappa'_c$ depends only on the bias $\mu$:

$$I_N(\infty) = \gamma'_c + \gamma'_s \langle A \rangle_z \stackrel{|\mu|\to\infty}{=} \frac{1}{2}\gamma_p \tanh\left(\frac{\mu}{2T}\right) \times \left(1 - \frac{\delta^2}{\delta_A^2}\right). \tag{101}$$

## 8 Conclusion

In this paper we have demonstrated how to fully exploit fermionic duality for the solution of open-system dynamics and transport right from the very first step of setting up the evolution and current equations. Unlike ordinary symmetry, this dissipative symmetry aids the construction of a *duality-adapted* orthogonal Liouville-space basis which is optimal for finding the distinct left *and* right eigenvectors of the master-equation rate superoperator $W$. The corresponding matrix elements have a highly constrained functional structure governed by *duality invariance*.

We developed these ideas concretely by deriving the full time-dependent solution (69)-(71) of the dynamics of a quantum dot proximized by a superconductor and weakly probed by charge- and heat currents into a normal-metal contact, a problem of high current interest. The obtained compact expressions capture a variety of effects when interaction and induced pairing compete, warranting a separate numerical analysis in Ref. [19]. Here we instead analytically investigated our all-encompassing solution and also discussed several covered limiting cases ($\alpha = 0$, $U = 0$, or $|\mu| \to \infty$) which may prove useful for comparison with other approaches.

For the general case we showed that the transient approach to the stationary state can be completely understood from duality-invariant decay rates and expectation values of two duality-adapted observables, parity and Andreev polarization, in the initial and *stationary* state of the system. By inspection of transition-rate expressions this is not obvious at all. Our result emerged by first expressing the solution as function of the stationary expectation values of these observables for the actual system *and* for its dual system with inverted energies. These stationary values provide the key to an exhaustive analysis of the solution [19] generalizing prior work [9] to include pairing. Moreover, we showed that the duality-based analysis can be extended to the measurable time-dependent transport quantities when including duality-invariant parts of the transport rates.

Likewise, extending other work [16], we combined duality with the detailed-balance property of the considered system [3,7]. We showed that the stationary values of observables in the *actual* model in fact determine their values of the *dual* model by a stationary duality relation [Eq. (59)]. This relation is "universal" for this class of system, independent of the values of all physical parameters: temperature $T$, coupling $\Gamma$, voltage $\mu$, but also the –attractive or repulsive– interaction $U$ and induced superconducting pairing $\alpha$. This naturally suggested a notion of *self-duality* of these observables which we showed occurs essentially whenever the interaction is irrelevant: This happens when interaction is either explicitly zero or made ineffective asymptotically and we illustrated these special cases.

When the interaction is relevant self-duality is violated but duality remains valid and is intimately linked with interactions within the open system. In particular, we showed that the *sign* of the interaction $U$ (repulsive/attractive) equals the sign of one of the duality invariants ($\gamma_C$), irrespective of the induced superconducting pairing $\alpha$. Likewise, the duality invariants clarified the singular connection for $\alpha \to 0$ to the problem of an interacting quantum dot plus normal metal without superconductor.

Our results underscore that fermionic duality is a powerful tool for advancing the quantitative understanding not only of the decay rates / time-scales of electronic open quantum systems but importantly also of the amplitudes which decide their (ir)relevance depending on the initial state. It is an intriguing open question how to systematically extend our work to even more complicated transport models (with orbital and spin splittings) within the broad class governed by weak-coupling fermionic duality. Also, for the present work, we have only used fermionic duality as a tool to simplify the procedure of solution and analysis of a *given* quantum master equation, here taken over from Ref. [36] for the infinite-gap limit. As a further step one can express duality already on the level of the derivation of the master equation, in particular, allowing for strong coupling effects to the normal reservoirs: the derivation of duality in Ref. [9] in fact applies to any parity conserving Hamiltonian which includes pairing terms describing infinite gap superconductors. A key remaining open question is thus whether for finite superconducting gap described by the approaches of Ref. [34] or [39] a more general duality relation can be found. The continued interest in (time-)controlled proximized nanoelectronic systems [60–64] provides a strong experimental impetus for such work.

# Acknowledgements

We thank A. Geresdi, F. Hassler, and J. Schulenborg for discussions.

**Funding information** L. C. O. acknowledges support by the Deutsche Forschungsgemeinschaft (RTG 1995). J.S. acknowledges financial support from the Swedish VR (project number 2018-05061) and the Knut and Alice Wallenberg Foundation.

# A Stationary duality and Kolmogorov detailed balance

Here we present two derivations of the stationary duality relation (59).

*Direct derivation:* Using Eqs. (43) we first express the parity $-1$ probability $\frac{1}{2}[1 - \langle p \rangle_z]$ in terms of $\langle A \rangle_z$ and $\langle A \rangle_{\bar{z}}$ by eliminating $(\frac{1}{2}\gamma_p + \gamma_C)/\gamma_p = [1 - \langle A \rangle_z / \langle A \rangle_{\bar{z}}]^{-1}$ and then express the remainder in $\langle A \rangle_z$. One then similarly expresses the parity $+1$ probability:

$$\frac{1}{2}\big[1 - \langle p \rangle_z\big] = \frac{1 - \langle A \rangle_z^2}{1 - \langle A \rangle_z / \langle A \rangle_{\bar{z}}}, \qquad \frac{1}{2}\big[1 + \langle p \rangle_z\big] = \frac{1 - \langle A \rangle_z \langle A \rangle_{\bar{z}}}{1 - \langle A \rangle_{\bar{z}} / \langle A \rangle_z}, \tag{A.1}$$

$$\frac{1}{2}\big[1 - \langle p \rangle_{\bar{z}}\big] = \frac{1 - \langle A \rangle_{\bar{z}}^2}{1 - \langle A \rangle_{\bar{z}} / \langle A \rangle_z}, \qquad \frac{1}{2}\big[1 + \langle p \rangle_{\bar{z}}\big] = \frac{1 - \langle A \rangle_z \langle A \rangle_{\bar{z}}}{1 - \langle A \rangle_z / \langle A \rangle_{\bar{z}}}. \tag{A.2}$$

The second row is obtained in a similar way and corresponds to formally replacing $z \to \bar{z}$ in the first row. From the first (second) relation in first (second) row one then finds the dual polarization (parity) in terms of the actual polarization and parity as given by Eq.(59a).

*Derivation using Eq. (74) of Ref. [16].* Fermionic duality was combined in Ref. [16] with the assumption that detailed balance $P_i/P_j = W_{ij}/W_{ji}$ holds, thus presupposing a unique stationary state with strictly positive probabilities and the validity of Kolmogorov condition [3]. These assumptions apply to the system considered here, see Ref. [16] for details. There it was shown that this implies a "universal" fermionic duality relation between the probabilities ($P_i$) of the energy eigenstates in the stationary state of the system and of the dual system ($\bar{P}_i$), separately counting possible degenerate states:

$$\bar{P}_i = \frac{P_i^{-1}}{\sum_k P_k^{-1}}. \tag{A.3}$$

This simple relation established an interesting connection to "relative stationary rareness" via Kac's lemma, see Ref. [16] for further discussion. In our notation these relations connect the energy-basis probabilities [Eq. (16)] of $|z\rangle = \sum_\tau z_\tau |\tau\rangle + z_1 |1\rangle$ and $|\bar{z}\rangle = \sum_\tau \bar{z}_\tau |\bar{\tau}\rangle + \bar{z}_1 |1\rangle$ of which we eliminate $z_1$ and $\bar{z}_1$ by normalization. Accounting for the degeneracy of the $N = 1$ spin states Eq. (A.3) gives for the Andreev states $\tau = \pm$

$$\bar{z}_\tau = \frac{(z_{-\tau})^{-1}}{\sum_\tau z_\tau^{-1} + 2(z_1/2)^{-1}} = \frac{z_1}{z_1 \sum_\tau z_\tau + 4 z_+ z_-} z_\tau = F z_\tau, \tag{A.4}$$

with rescaling factor (59b)

$$F = \frac{z_1}{z_1 \sum_\tau z_\tau + 4 z_+ z_-} = \frac{\frac{1}{2}\big[1 - \langle p \rangle_z\big]}{\frac{1}{2}\big[1 + \langle p \rangle_z\big] - \langle A \rangle_z^2}. \tag{A.5}$$

Changing to duality-adapted variables $z_\tau = \frac{1}{4}\big[1 + \langle p \rangle_z\big] + \tau \frac{1}{2}\langle A \rangle_z$ and $z_1 = \frac{1}{2}\big[1 - \langle p \rangle_z\big]$, we obtain relation (59a)

$$\langle \bar{A} \rangle_{\bar{z}} \equiv \sum_\tau \tau \bar{z}_\tau = F \sum_\tau \tau z_\tau = F \langle A \rangle_z, \tag{A.6}$$

$$\frac{1}{2}\big[1 + \langle p \rangle_{\bar{z}}\big] \equiv \sum_\tau \bar{z}_\tau = F \sum_\tau z_\tau = F \frac{1}{2}\big[1 + \langle p \rangle_z\big]. \tag{A.7}$$

Although this derivation highlights the importance of detailed balance for the "universality" of this relation, it still relies crucially on fermionic duality through Eq. (A.3).

# B Condition for self-duality

Here we consider the self-duality condition $\gamma_c = \frac{1}{2}\gamma_p$ for the charge decay rate [Eq. (11),(49a)]:

$$\gamma_c = \frac{1}{2}\sum_{\eta\tau} W_{1,\tau}^{\eta} = \frac{1}{2}\gamma_p \sum_{\lambda}\frac{1}{2}\Big(1 + \lambda\frac{\delta}{\delta_A}\Big)\sum_{\eta} f^{-\eta}\Big(\frac{1}{2}(\lambda\delta_A + \eta U) - \mu\Big). \tag{B.1}$$

## B.1 Exact self-duality and its breakdown due to interaction

The strict lower / upper bounds (63), $\gamma_c \gtrless \frac{1}{2}\gamma_p$ for $U \gtrless 0$ at finite temperature $T$ follow using $f(x) = [e^{x/T} + 1]^{-1} = \frac{1}{2}[1 + (1 - e^{x/T})/(1 + e^{x/T})]$ and $f^{\eta}(x) = f(\eta x)$ :

$$\sum_{\eta} f^{-\eta}(x + \eta y) = 1 + \frac{1 - e^{-2y/T}}{(e^{-(x+y)/T} + 1)(e^{(x-y)/T} + 1)} \gtrless 1. \tag{B.2}$$

With $x = \lambda\delta_A - \mu$ and $y = U/2$, Eq. (B.1 ) gives $\gamma_c \gtrless \frac{1}{2}\gamma_p \sum_{\lambda}\frac{1}{2}(1 + \lambda\frac{\delta}{\delta_A}) = \frac{1}{2}\gamma_p$ for $U \gtrless 0$. For $T < \infty$ the condition for self-duality (60), $\gamma_c = \gamma_p/2$ [Eq. (62)], holds for all values of parameters other than $U$ if and only if $U = 0$ since Eq. (B.2 ) is an equality if and only if $y = 0$. For $T \to \infty$ equality is satisfied trivially for all parameter values.

## B.2 Asymptotic self-duality

It is possible to achieve self-duality asymptotically, i.e., by making one energy scale dominate all others (excluding $\Gamma$ which cancels out $\gamma_c = \gamma_p/2$). All that is needed to eliminate the dependence on $\eta$ in the argument of the Fermi distribution in Eq. (B.1 ). This is always possible by taking high temperature $T \gg |U|, |\alpha|, |\delta|$

$$\gamma_c \approx \frac{1}{2}\gamma_p \sum_{\lambda}\frac{1}{2}\Big(1 + \lambda\frac{\delta}{\delta_A}\Big)\sum_{\eta} f^{-\eta}(0) = \frac{1}{2}\gamma_p. \tag{B.3}$$

This "trivial" limit in fact plays a crucial role in the derivation of fermionic duality [9, 10], in the underlying renormalized perturbation theory [18] and methods based on this [17, 65, 66]. If temperature is not dominant, to achieve self-duality one needs to make ineffective the $\eta$ dependence which is tied to the interaction $U$ in Eq. (B.1 ) in the transition energy $E_{\eta,\eta\lambda} = \frac{1}{2}(\lambda\delta_A + \eta U) - \mu$. For large bias voltage $|\mu| \gg |\delta|, |\alpha|, U$ such that $|\mu| \gg |E_{\eta,\eta\lambda}|$ discussed in Sec. 7.3 this works:

$$\gamma_c \approx \frac{1}{2}\gamma_p \sum_{\lambda}\frac{1}{2}\Big(1 + \lambda\frac{\delta}{\delta_A}\Big)\sum_{\eta} f^{-\eta}(-\mu) = \frac{1}{2}\gamma_p. \tag{B.4}$$

Similarly, for large pairing $|\alpha| \gg |\delta|$ relative to detuning we recover the $U \to 0$ limit discussed in Sec. 7.2:

$$\gamma_c \approx \frac{1}{2}\gamma_p \sum_{\lambda}\frac{1}{2}\Big(1 + \lambda\frac{\delta}{\alpha}\Big)\sum_{\eta} f^{-\eta}\big(\lambda\frac{1}{2}|\alpha| - \mu\big) = \frac{1}{2}\gamma_p. \tag{B.5}$$

For large detuning $|\delta| \gg |\alpha|, U$ one recovers

$$\gamma_c \approx \frac{1}{2}\gamma_p \sum_{\lambda}\frac{1}{2}\Big(1 + \lambda\frac{\delta}{|\delta|}\Big)\sum_{\eta} f^{-\eta}\big(\lambda\frac{1}{2}|\delta| - \mu\big) = \frac{1}{2}\gamma_p, \tag{B.6}$$

which we discussed in Sec. 7.1 for $|\delta| \neq 0$ taking $\alpha \to 0$ and then $U \to 0$.

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
