# Peer review of "Solution of master equations by fermionic-duality: Time-dependent charge and heat currents through an interacting quantum dot proximized by a superconductor"

_SciPost Physics, doi:SciPost Phys. 14, 095 (2023)_

## Round 1 · Referee Report · Anonymous (Referee 1) · 2022-11-14

Report

In the manuscript, the Authors demonstrate how to construct the fully analytical solution to an interacting quantum transport problem guided by the fermionic duality symmetry. The latter is a symmetry of fermionic open quantum systems previously explored by (part of) the Authors, see Ref.[8,9] of the manuscript, which includes the PT symmetry of the (Markovian) Lindblad dynamics as a special case and is valid in the presence of strong coupling/non-Markovianity.

The system used for the demonstration is an interacting spinfull single-level quantum dot weakly connected to a wide-band metallic lead and proximized by a supeconductor with infinitely large superconducting gap.
By exploiting the fermionic duality of the system, the transient energy and charge currents are expressed in terms of stationary expetaction values of operators such as the dot polarization, and the elements of the rate superoperator expressed in a suitable basis, whose choice is enforced by the duality symmetry.

The approach is useful and interesting, the manuscript very well written, and the exposition clear and detailed. The derivations are sound, no flaws are apparent to me. For these reasons the manuscript deserves in my opinion publication in : SciPost Physics.

I have a few minor comments

  • The general conditions under which the fermionic duality holds could be stated for completeness

  • The Authors should comment on the practical limitations of the approach in general and in the specific example considered here (e.g. finite gap, non-degenerate case)

  • A quantum dot connected to superconducting leads in the case of nonequilibrium, finite gap, and AC drive is addressed in the recent article J. Siegl 2022, arXiv:2205.13936

  • validity: high
  • significance: -
  • originality: -
  • clarity: top
  • formatting: excellent
  • grammar: excellent

Author:  Maarten Wegewijs  on 2022-12-21  [id 3170]

(in reply to Report 1 on 2022-11-14)

We thank the referee for the positive appraisal of the work, in particular of the full exposition.

Response to the referee's suggestions:

The general conditions under which the fermionic duality holds could be stated for completeness

Revision A brief overview of the class of open systems, which exhibit a fermionic duality, is now given in the introduction Section 1. It now clearly mentions the general form and conditions of the fermionic duality, referring to prior works going beyond the weak-coupling assumption made here. Section 4 now clearly refers to this more general framework.

The Authors should comment on the practical limitations of the approach in general and in the specific example considered here (e.g. finite gap, non-degenerate case)

Response In Section 2 we already mention the limitations of the applicability of our specific discussion.

Revision In the revised conclusion, Section 8, we point out our limited use of fermionic duality, namely only as a tool to simplify the solution and analysis of a given quantum master equation (In our case taken from Sothman et al in Ref. 8). We highlight that for finite-gap superconducting reservoirs duality has so far not been exploited to simplify the preliminary task of computing the memory kernel from the underlying microscopic model as it has been by Schulenborg et al in the supplement of PRB 93, 081411(R) (2016).

A quantum dot connected to superconducting leads in the case of nonequilibrium, finite gap, and AC drive is addressed in the recent article J. Siegl 2022, arXiv:2205.13936

Revision We appreciate this interesting reference, it is now cited in the revised introduction, Section 1, and summary, Section 8, where we address the referee's previous question regarding the limitations of the approach.

---

## Round 1 · Referee Report · Anonymous (Referee 2) · 2022-11-25

Strengths

1- It presents a careful description of the method, including all details 2- If presents a powerful and innovative method for the computation of transient transport in quantum dot devices

Weaknesses

1- (maybe contradictory with strength #1): the manuscript is very long 2- the introduction is difficult to read for non experts.

Report

The manuscript "Solution of master equations by fermionic-duality: Time-dependent charge and heat currents through an interacting quantum dot proximized by a superconductor" presents a detailed analysis of the transient dynamics in a device composed of a proximized single-ĺevel quantum dot (by being coupled to a infinite-gap superconductor) tunnel coupled to a normal metal contact. This is neither an unexplored configuration neither a new problem (relevant references are cited), however the authors introduce a powerful method that considerably simplifies its solution, adding important insights on the relevance of fermionic symmetries. The method is also not totally new :the authors have published a number of references using it in simple (simpler) configurations, but it is shown here to be useful for more complex situations, at the same time including new and interesting results in the form of constraints of the dual quantities and the universal and simple relation between their stationary expected values. The text is "in average" clearly written, with all the details of the method carefully exposed. For these reasons, I consider it meets the acceptance criteria to be published in SciPost.

Below, I include a list of more detailed comments/suggestions that may be considered:

  • The results are kept on the formal level, with numerical results being referred to a "manuscript in preparation" by the same authors (ref. [18]). Being unnecessary to make the manuscript longer, I wonder if it would increase its presentation by plotting some results (e.g., a comparison of the limiting cases discussed in section 7).

  • As I said, the manuscript is carefully and "in average" clearly written . With this, I mean that most of it is very clear and easy to follow, especially in what concerns the derivation of the results, what will be very useful for the interested readers. Unfortunately, this is not the case for the introduction: it is written in a very precise way which I'm afraid is only accessible for specialists or readers that have already known about fermionic duality before. For instance, most of it refers to duality, and the advantages of using "duality-dictated observables" without the reader clearly knowing what duality actually is. Other concepts like the difference between "transition" and "transport rates", or "a shifted version of the transition rate matrix" are not totally transparent. They become clearer as the text advances, but are confusing in the introduction, I wonder if some of those more technical discussions could be more convenient in the conclusions section than before the concepts are presented. I would encourage the authors to try to make the introduction a bit easier to read for non experts.

  • In my opinion, the main motivation of the manuscript is to present the strengths of the duality-based treatment of the transient dynamics of quantum dot systems. In that case, it would be useful to know better about its limitations. For instance, a sequential tunneling rate equation is used. Can similar arguments apply to higher order expansions (cotunneling), other master equations (Lindblad, Redfield...)? A wide-band limit seems to have been assumed. Is the duality affected if tunneling rates depend on energy?

  • Rates $\gamma_C$ and $\gamma_S$ turn out to have a physical interpretation but they are introduced as an ansatz in eq. (29). I wonder whether this treatment can be generalized to (even) more complex configurations where additional rates may be relevant and revealed this way, or does one need to have a previous intuition/knowledge of which they are?

Minor things: - page 10, first paragraph of section 5: is is said that rate variables "can occur by their own" what does this mean? - footnote 6: there is a ·the" too much ("...but our choice ensures...") - below eq. (43), doesn't one need to transform $\gamma_s\rightarrow-\gamma_s$ as well?

  • validity: top
  • significance: good
  • originality: high
  • clarity: good
  • formatting: good
  • grammar: perfect

Author:  Maarten Wegewijs  on 2022-12-21  [id 3171]

(in reply to Report 2 on 2022-11-25)

We thank the referee for careful reading of the manuscript -despite its admitted considerable length- and the positive appraisal.

Being unnecessary to make the manuscript longer, I wonder if it would increase its presentation by plotting some results (e.g., a comparison of the limiting cases discussed in section 7).

Response We have long considered this possibility but have decided against it since any example plot of dynamics requires an initial condition which is experimentally well-motivated and consistent with the approximations in Section 2. Introducing this requires quite some care and involves novel results which cannot be briefly conveyed. They are presented in the cited follow up work [Ref. 19 of revised manuscript] with the application and numerical analysis of all regimes that our concrete analytical result covers.

Revision Concrete examples the limiting cases which the referee suggests were reported already in the cited works, in particular Schulenborg etal in Phys. Rev. B 93, 081411(R) (2016) and Appl. Phys. Lett. 116, 243103 (2020). After Eq. (89) we now refer more clearly to these works for tangible plots and discussion.

the introduction: it is written in a very precise way which I'm afraid is only accessible for specialists or readers that have already known about fermionic duality before.

Revision We thank the referee for this useful feedback. We have revised and shortened the introduction with the non-specialist reader in mind.

For instance, most of it refers to duality, and the advantages of using "duality-dictated observables" without the reader clearly knowing what duality actually is.

Revision We now introduced duality immediately from the beginning and furthermore decided to not introduce the concept of duality-dictated observables in the introduction.

Other concepts like the difference between "transition" and "transport rates",

Revision We have clarified the difference between the two types of rates.

or "a shifted version of the transition rate matrix" are not totally transparent. They become clearer as the text advances, but are confusing in the introduction, I wonder if some of those more technical discussions could be more convenient in the conclusions section than before the concepts are presented.

Revision In line with the referee's suggestion we have removed this paragraph and merged it with the conclusion Section 8.

it would be useful to know better about its limitations. For instance, a sequential tunneling rate equation is used. Can similar arguments apply to higher order expansions (cotunneling), other master equations (Lindblad, Redfield...)?

Revision These were mentioned but the revised introduction now highlights these more clearly.

A wide-band limit seems to have been assumed. Is the duality affected if tunneling rates depend on energy?

Revision For weak tunneling the duality can be extended to energy-dependent spectral density see Schulenborg, PRB 98, 235405 (2018) where it was applied charge and heat transport with normal conducting reservoirs. We now mention this more prominently in the introduction Section 1.

It is good that the referee brings up this point: The derivation of duality presented in the above work in fact also applies to proximized quantum dot Hamiltonians (1) in our present study and this is easily overlooked. It is highlighted now in Section 1 and 8 (also in response to Referee 1).

Rates ${\gamma_{C}}$ and ${\gamma_{S}}$ turn out to have a physical interpretation but they are introduced as an ansatz in eq. (29). I wonder whether this treatment can be generalized to (even) more complex configurations where additional rates may be relevant and revealed this way, or does one need to have a previous intuition/knowledge of which they are?

Response The duality mapping implies a general formal strategy for constructing invariant quantities from linear combinations of the transition rates (instead of mapping one problem onto the other as in previous works). Some of the invariant quantities will have more tangible physical meaning (e.g. as decay rates i.e. eigenvalues of the rate matrix) whereas others characterize the various asymmetries between the various transition rates. This works in general but the details may vary with the complexity of the system.

-For weak coupling it can be applied separately to each electrode that independently adds a term to the master equation rate matrix. For normal electrodes this amounts to collecting the symmetric and antisymmetric part of fermi-functions which is easy as illustrated in Schulenborg B 93, 081411(R) (2016).

-In the present paper the duality invariant rates were constructed to also cover the transport equations. As a most general ansatz, we chose them to be (anti-) symmetric with respect to inversion of the particle index ($\eta$) or the Andreev state index ($\tau$) or both. The particle index $\eta$ can be expected to play a role in these kernel rate combinations, as it naturally occurs in the total charge current. The other Andreev state index $\tau$, however, also occurs in the current kernel rates, when we explicitly account for a transient displacement current into the quantum dot. Only in the stationary limit, this displacement current vanishes due to charge conservation. Therefore we expect the Andreev state index $\tau$ also plays a role.

Finally, we note that we here shown how to construct the invariants after using the standard derivation of the rate matrix. To automatically obtain the rates in duality-adapted form should be possible using the approach of Schulenborg, Saptsov et al presented in the supplement of B 93, 081411(R) (2016) which applies also to infinite-gap superconducting leads. This is interesting since it extends beyond the leading order in the coupling to the reservoirs. It is interesting whether for finite gap a similar result can be found. This is now mentioned in the summary Section 8 (also in response to referee 1).

page 10, first paragraph of section 5: it is said that rate variables "can occur by their own" what does this mean?

Response It means that the invariance of the rate variables ensures that the corresponding cross-related terms of the kernel (as required by the duality relation (21)) are represented by one and the same rate, which enables a most compact representation.

Revision We have removed this sentence since it apparently confuses more than it clarifies at this point in the text.

below eq. (43), doesn't one need to transform $\gamma_S \to - \gamma_S$ as well?

Revision This is a good point: in fact in Eq. (43) one should transform $\gamma_s$ but with the + sign, $\gamma_s \to + \gamma_s$ [Eq. (30)]. This is now pointed out.

---

## Round 2 · Referee Report · Anonymous (Referee 2) · 2022-12-22

Report

The authors response is clarifying and the modifications made to the introduction and conclusions in the revised version of the manuscript greatly increases its readability. Hence, I recommend it for publication without further requests.

---

## Round 2 · Referee Report · Anonymous (Referee 1) · 2023-1-9

Report

The revised version of the manuscript presents some improvements, in terms of readability and completeness, upon the already fine first version. For this reason I recommend publication in SciPost Physics.

---

## Round 2 · List of Changes

1. Introduction was shortened + revised to improve accessibility for non-expert reader

  2. After Eq. (22): relation to general duality pointed out.

  3. After Eq. (43b): a missing substitution mentioned.

  4. After Eq. (89): references to earlier worked-out examples highlighted.

  5. Summary: material cut from introduction merged + outlook paragraph extended.

  6. References [8] and [39] were added

---

## Editorial Decision

published